# MULTIMODAL DATASET DISTILLATION VIA PHASED TEACHER MODELS

**Shengbin Guo**[1]* , **Hang Zhao**[1]*, **Senqiao Yang**[2], **Chenyang Jiang**[1]
**Yuhang Cheng**[1], **Xiangru Peng**[1], **Rui Shao**[1], **Zhuotao Tian**[1]†
[1]Harbin Institute of Technology, Shenzhen    [2]The Chinese University of Hong Kong
`shengbinguo2022@gmail.com, 200110431@stu.hit.edu.cn`

## ABSTRACT

Multimodal dataset distillation aims to construct compact synthetic datasets that enable efficient compression and knowledge transfer from large-scale image-text data. However, existing approaches often fail to capture the complex, dynamically evolving knowledge embedded in the later training stages of teacher models. This limitation leads to degraded student performance and compromises the quality of the distilled data. To address critical challenges such as pronounced cross-stage performance gaps and unstable teacher trajectories, we propose Phased Teacher Model with Shortcut Trajectory (PTM-ST)—a novel phased distillation framework. PTM-ST leverages stage-aware teacher modeling and a shortcut-based trajectory construction strategy to accurately fit the teacher's learning dynamics across distinct training phases. This enhances both the stability and expressiveness of the distillation process. Through theoretical analysis and comprehensive experiments, we show that PTM-ST significantly mitigates optimization oscillations and inter-phase knowledge gaps, while also reducing storage overhead. Our method consistently surpasses state-of-the-art baselines on Flickr30k and COCO, achieving up to 13.5% absolute improvement and an average gain of 9.53% on Flickr30k. Code: `https://github.com/Previsior/PTM-ST`.

## 1 INTRODUCTION

Large-scale multimodal models such as CLIP (Radford et al., 2021), BLIP (Li et al., 2022), and Flamingo (Alayrac et al., 2022) have established image-text pairs as a fundamental aspect of visual-language pre-training (VLP). Their successors, e.g., BLIP-2 (Li et al., 2023) and LLaVA (Liu et al., 2023), further extend this paradigm to multimodal large language models (MLLMs) (Jia et al., 2021; Chen et al., 2023c). However, the application of such extensive multimodal data incurs substantial storage as well as computational overhead (Hoffmann et al., 2022; Kang et al., 2024), highlighting an urgent need for efficient data utilization and improved model training efficiency.

Dataset Distillation (DD) (Wang et al., 2018; Lei & Tao, 2023) offers a promising solution to this challenge by synthesizing a compact surrogate dataset that compresses the dataset by orders of magnitude without hurting performance (Zolfaghari et al., 2021; Lin & Hu, 2022). Despite notable progress in unimodal tasks across text (Li & Li, 2021), video (Wang et al., 2024) and graph (Xu et al., 2023) domains, directly extend DD to multimodal scenarios presents considerable difficulties.

Although a few recent studies (e.g., MTT-VLL (Wu et al., 2023) and LoRS (Xu et al., 2024)) have attempted to extend dataset distillation methods to multimodal tasks, these approaches primarily focus on directly adapting existing unimodal distillation strategies to multimodal scenarios or enhancing cross-modal alignment through the incorporation of low-rank similarity matrices. Overall, such methods largely remain at a superficial level, emphasizing the design of data structures or distance metrics, and concentrate mainly on representation and alignment challenges in multimodal learning (Du et al., 2023; 2024). However, they fall short of investigating the underlying mechanistic differences between multimodal and unimodal settings in the context of dataset distillation, differences that likely govern the stability, efficiency, and ultimate effectiveness of any practical

---

*Equal contribution. †Corresponding author: `tianzhuotao@hit.edu.cn`.

distillation strategy. This gives rise to a fundamental scientific question: *What are the essential distinctions between multimodal and unimodal dataset distillation tasks and how should we improve?*

**Key Observation.** To explore the fundamental distinctions between multimodal and unimodal dataset distillation, we conduct a comparative study under a unified MTT framework. Our results reveal that, unlike unimodal tasks—where teacher guidance remains effective throughout the whole training process—multimodal distillation benefits primarily from the early 20–30% of training epochs. As shown in Figure 2 (a), although teacher model continues to improve in later stages, using it directly for distillation leads to a sharp drop in student performance.

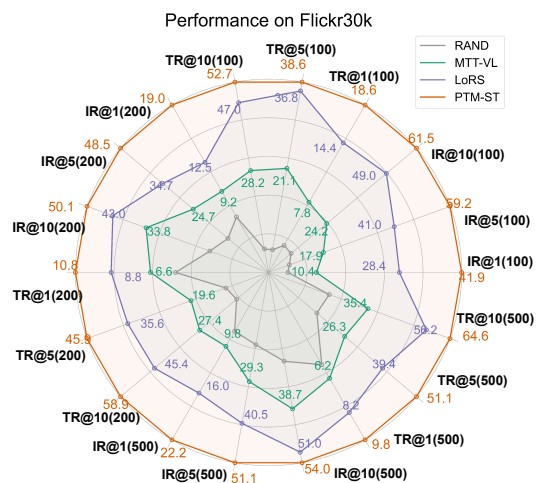

Figure 1: PTM-ST has achieved far superior performance than other selection or distillation methods in different metrics on Flickr30k.

We hypothesize that this limitation stems from the inherent sparsity of multimodal data and the absence of explicit semantic constraints, which lead the teacher model to encode substantially different forms of knowledge at different training stages. As a result, existing distillation methods struggle to effectively leverage the knowledge embedded in the teacher model across stages to support the continual learning of the student model. These findings underscore the necessity of developing a distillation framework that can dynamically adapt to the evolving knowledge representation of the teacher model, thereby enabling the student to absorb intermediate signals more effectively, mitigate representation drift, and achieve stable, continual improvements in downstream tasks.

**Our Solution.** To address the limitations of conventional distillation strategies in modeling the evolving dynamics of teacher models, we propose a novel framework: **P**hased **T**eacher **M**odeling with **S**hortcut **T**rajectory (**PTM-ST**). Built upon the dual-loop architecture of the original MTT framework, PTM-ST introduces two core components: (1) Phased Teacher Modeling (PTM), which decomposes the overall distillation objective into a sequence of sub-tasks, each associated with a phase-specific teacher model that adapts to the semantic evolution across training stages; and (2) Shortcut Trajectory (ST), which constructs a smoothed and stabilized trajectory to guide student learning, mitigating the instability and noise often present in late-stage teacher updates.

By capturing stage-specific knowledge more precisely and ensuring stable knowledge transfer, PTM-ST significantly improves the quality of distilled data and facilitates more effective student training. As shown in Figure 1, we conducted extensive experiments on the Flickr30K (Plummer et al., 2015) and MS-COCO (Lin et al., 2014) datasets, validating the superior performance and robustness of our method compared to existing state-of-the-art approaches.

To summarize, our contributions are as follows:

- We identify and analyze the phased knowledge gap in multimodal dataset distillation, validating the phenomenon through visualization and supporting theoretical analysis.

- We propose the PTM-ST framework, combining a phased distillation strategy with trajectory endpoint matching, significantly improving distillation stability and performance.

- Our method outperforms existing techniques in multimodal image-text retrieval distillation, pushing the boundaries of multimodal distillation research and its applications.

## 2 PRELIMINARIES

In this section, we briefly review foundational concepts relevant to Section 2.1 to establish background for our work. Then, Section 2.2 details our key insights and design motivations.

### 2.1 BACKGROUND

**Multimodal Dataset Distillation.** Dataset distillation seeks to synthesize a compact, representative subset that effectively approximates the statistical properties (Wang et al., 2022; 2025c; Zhao & Bilen, 2023; Lee et al., 2022; Deng & Russakovsky, 2022) and training dynamics (Chen et al., 2023b; Du et al., 2024; Shao et al., 2024a; Yin et al., 2023) of large-scale datasets. Multimodal dataset distillation (MDD) extends this concept from unimodal data (e.g., images) to multimodal data (e.g., image-text pairs). Current approaches predominantly fall into meta-learning-based (Zhou et al., 2022) and distribution-matching-based (Lee & Chung, 2024) strategies. While these methods perform well on unimodal benchmarks (Lei et al., 2024), their extension to multimodal settings remains challenging due to the intrinsic complexity and heterogeneity of multimodal data (Chen et al., 2023a; Peng et al., 2024; Aljundi et al., 2019b;a), requiring further exploration.

**The Baseline.** We adopt LoRS (Xu et al., 2024) as the baseline for MDD, which is based on the Match-Training-Trajectory (MTT) framework (Cazenavette et al., 2022; Wu et al., 2023; Cui et al., 2023b). MTT optimizes a synthetic dataset $\tilde{\mathcal{D}}$ by aligning the parameter trajectories of models trained separately on the synthetic data and the real dataset $\mathcal{D}$.

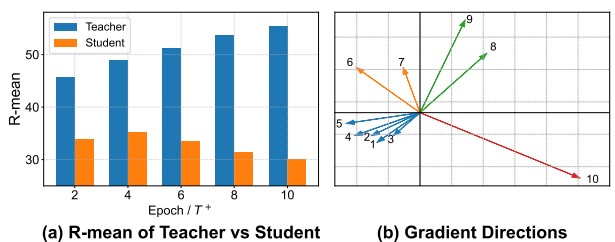

(a) R-mean of Teacher vs Student   (b) Gradient Directions

Figure 2: **(a)** shows that using a late-stage, high-performing teacher during distillation leads to a decline in student performance. **(b)** shows the directions of different matching range gradients (after PCA dimensionality reduction).

Concretely, a teacher model is trained on $\mathcal{D}$ for $n$ epochs, producing a trajectory $\tau = \{\theta_0, \theta_1, \ldots, \theta_n\}$ that captures the parameter evolution. Let $T^-$ be the lower bound and $T^+$ be the upper bound on the sampling range of $T$, where $T^-$ is generally set to 0. The student model is then trained on synthetic dataset $\tilde{\mathcal{D}}$ to match the teacher's trajectory from $\theta_T$, minimizing the discrepancy between their parameter states after additional training steps. Formally, the objective is to minimize:

$$\mathcal{L}_{MTT}(\tilde{\mathcal{D}}, \theta_T) = \frac{||\tilde{\theta}_{T+t} - \theta_{T+\Delta T}||_2^2}{||\theta_T - \theta_{T+\Delta T}||_2^2}, \tag{1}$$

where $\tilde{\theta}_{T+t}$ denotes the model parameters obtained by training $t$ steps on $\tilde{\mathcal{D}}$ starting from $\theta_T$, and $\Delta T$ represents the length of matching range in $\tau$. Building upon this foundation, LoRS applied an augmentation technique to the synthetic dataset to enhance its expressive capability. More details about previous work can be found in Appendix A.

### 2.2 MOTIVATION

**Knowledge gap across learning stages.** Despite recent progress in multimodal dataset distillation (MDD), existing methods predominantly rely on the early-stage checkpoints of the teacher model—typically within the first 20–30% of training epochs ($T^+ \leq 0.3n$). However, as shown in Figure 2 (a), while the teacher's performance continues to improve in later epochs, directly incorporating these late-stage checkpoints often leads to degraded student performance. This suggests that *knowledge discrepancies emerge across different training stages*, and the student model struggles to assimilate the teacher's late-stage knowledge effectively, compromising the distillation results.

To further investigate this phenomenon, inspired by DDGM (Zhao et al., 2020), we analyze the gradients propagated from the teacher to the synthetic dataset at various epochs. As illustrated

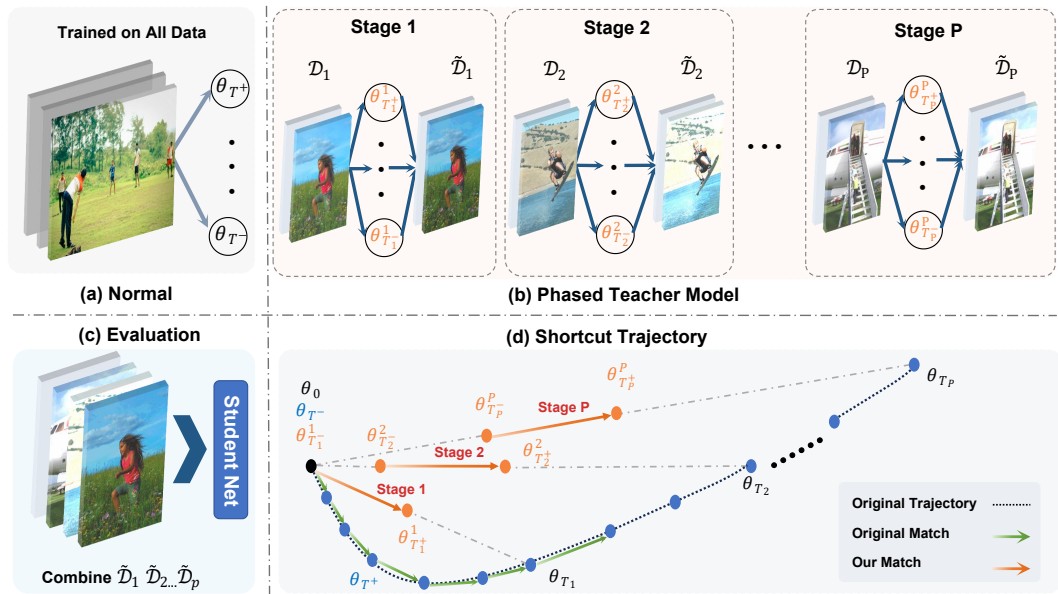

Figure 3: **(a)** shows the conventional single-stage training with a fixed teacher model and uniform data use. In contrast, **(b)** depicts our Phased Teacher Model (PTM), which employs different teacher models across multiple training stages to distill knowledge to specific data subsets. **(c)** illustrates the aggregation of all distilled subsets for final student evaluation. Additionally, **(d)** presents our Shortcut Trajectory (ST) strategy that dynamically generates stage-adaptive teacher models, improving distillation effectiveness and robustness.

in Figure 2 (b), the gradient norm increases progressively throughout training, implying intensified update signals. However, the gradient directions exhibit high variability and inconsistency, revealing substantial instability in the learning signals over time. These observations highlight two critical challenges: pronounced stage-wise knowledge shifts and unstable optimization dynamics in later training phases—both of which hinder effective knowledge transfer in current MDD frameworks.

**Limitations of existing methods.** Existing multimodal dataset distillation (MDD) approaches (Xu et al., 2024) typically rely on a single synthetic subset, which subjects the distilled dataset to persistently high-magnitude and unstable gradient updates. Such instability compromises the consistency and reliability of knowledge transfer, ultimately exacerbating the performance gap between teacher and student models. Moreover, recent findings (Guo et al., 2023; Chen et al., 2023b) highlight that the generalization capability of deep models emerges from their progressive assimilation of training samples across different learning stages. This suggests that conventional single-subset MDD strategies are fundamentally constrained in their ability to capture the full spectrum of knowledge distributed throughout the training process. For more analysis, see Appendix B.1.

Motivated by these insights, we propose a phased distillation mechanism that enables the student model to incrementally align with the teacher's knowledge trajectory across successive learning stages. By decomposing the distillation process into multiple temporal phases, our approach facilitates more stable gradient updates and improves the fidelity of knowledge transfer from the original dataset to the synthetic one, therefore enhancing the distillation effect.

## 3 METHOD

This section introduces the Phased Teacher Model (PTM) with Shortcut Trajectory (ST), an approach designed to address the challenge highlighted in Section 2.2. The overall framework is illustrated in Figure 3. Section 3.1 show the design of PTM, followed by the ST strategy in Section 3.2.

---

**Algorithm 1** Distillation: Phased Teacher Model with Shortcut-Trajectory

---

**Input:** Real dataset $\mathcal{X}, \mathcal{Y}$. Size, iteration, matching range, interpolation endpoint of each subset: $\{(N_p, I_p, T_p^-, T_p^+, t_p)\}_{p=1}^P$. Decay parameter $\alpha$, learning rate $\gamma$. Synthesis steps $t$, expert epochs $\Delta T$.
**Output:** A series of synthetic datasets: $\tilde{\mathcal{D}}_1, \tilde{\mathcal{D}}_2, \ldots, \tilde{\mathcal{D}}_P$.

1: **Generating synthetic subsets:**
2: **for** $p = 1, 2, \ldots, P$ **do**
3:     Initialize $\tilde{\mathcal{X}}_p, \tilde{\mathcal{Y}}_p$ by real data, $\tilde{S}_p = \mathbb{I}_{N_p}$ (identity matrix). $\hat{\mathcal{D}}_p^0 = (\tilde{\mathcal{X}}_p, \tilde{\mathcal{Y}}_p, \tilde{S}_p)$
4:     Calculate Shortcut-Trajectory $\{\theta_{T_p^-}^p, \ldots, \theta_{T_p^+}^p\}$ based on $\theta_0$ and $\theta_{t_p}$
5:     **for** $i = 1, 2, \ldots, I_p$ **do**
6:         Sample an initial network parameter $\theta_T^p$ in $\{\theta_{T_p^-}^p, \ldots, \theta_{T_p^+}^p\}$
7:         Train the network for $t$ steps on $\tilde{\mathcal{D}}_p$ to $\tilde{\theta}_T^p$
8:         Compute MTT loss $\mathcal{L}_{\text{MTT}} = \|\tilde{\theta}_T^p - \theta_{T+\Delta T}^p\|^2 / \|\theta_T^p - \theta_{T+\Delta T}^p\|^2$
9:         Gradient descent: $\tilde{\mathcal{X}}_p \leftarrow \tilde{\mathcal{X}}_p - \gamma_{\mathcal{X}} \nabla_{\tilde{\mathcal{X}}_p} \mathcal{L}_{\text{MTT}}, \tilde{\mathcal{Y}}_p \leftarrow \tilde{\mathcal{Y}}_p - \gamma_{\mathcal{Y}} \nabla_{\tilde{\mathcal{Y}}_p} \mathcal{L}_{\text{MTT}},$
          $\tilde{S}_p \leftarrow \tilde{S}_p - \gamma_S \nabla_{\tilde{S}_p} \mathcal{L}_{\text{MTT}}$
10:        EMA: $\tilde{\mathcal{D}}_p^i \leftarrow (\tilde{\mathcal{X}}_p, \tilde{\mathcal{Y}}_p, \tilde{S}_p), \hat{\mathcal{D}}_p^i \leftarrow \alpha \hat{\mathcal{D}}_p^{i-1} + (1 - \alpha) \tilde{\mathcal{D}}_p^i$
11:    **end for**
12:    $\tilde{\mathcal{D}}_p \leftarrow \hat{\mathcal{D}}_p^{I_p}$
13: **end for**

---

## 3.1 PHASED TEACHER MODEL

As mentioned in the Section 2.2, simply switching the teacher model according to the training phase does not yield satisfactory results. To better capture stage-dependent learning dynamics and provide smoother guidance, we propose the Phased Teacher Model (PTM). This strategy progressively introduces different teacher models during training, allowing the distilled dataset to absorb as much information as possible from different teacher models at each stage.

Specifically, as shown in Figure 3 (b), we divide the distillation process into P stages, with each stage independently distilling a small subset $\tilde{\mathcal{D}}_p$, where $p = 1, 2, ..., P$ denotes the stage id.

In our method, we dynamically adjust the sampling range of the trajectory matching starting point. The sampling range for stage $p$ is $\{T_p^-, ..., T_p^+\}$, and this range evolves as training progresses. This means that, at each stage, we use a different teacher model to guide the update of the distillation dataset, allowing each subset to focus on distilling different aspects of knowledge. As analysed in Section 4.2, the union of all subsets, $\tilde{\mathcal{D}}_1 \cup \cdots \cup \tilde{\mathcal{D}}_P$, can thus capture the complete training dynamics.

During testing, we adopt a progressive training scheme. Specifically, the model is first trained on $\tilde{\mathcal{D}}_1$, then on $\tilde{\mathcal{D}}_2$, and so on. This corresponds to the different stages of our distillation process, allowing students to gradually extract the teacher model's training dynamics.

Formally, We generated the dataset for stage $p$ as follows:

$$\tilde{\mathcal{D}}_p^* = \arg\min_{\tilde{\mathcal{D}}_p} \mathbb{E}_{T \sim (T_p^-, ...T_p^+)} \mathcal{L}_{PTM}(\tilde{\mathcal{D}}_p, \theta_T^p), \tag{2}$$

where $\mathcal{L}_{PTM}$ denotes the trajectory matching loss defined in Eq. 3:

$$\mathcal{L}_{PTM}(\tilde{\mathcal{D}}_p, \theta_T^p) = \frac{\|\tilde{\theta}_{T+t}^p - \theta_{T+\Delta T}^p\|_2^2}{\|\theta_T^p - \theta_{T+\Delta T}^p\|_2^2} \quad s.t. \ \tilde{\theta}_{T+t}^p = \arg\min_\theta l(\tilde{\mathcal{D}}_p; \theta_T^p), \tag{3}$$

where $l(\tilde{\mathcal{D}}; \theta)$ denotes the loss of the student model trained on the distillation dataset $\tilde{\mathcal{D}}$ with parameter $\theta$, and $\theta_T^p$ denotes the updated teacher model parameters generated according to the Shortcut Trajectory (ST) strategy, which will be elaborated in the subsequent section.

## 3.2 SHORTCUT TRAJECTORY

PTM enables the distilled dataset to capture multi-stage knowledge throughout the training process, yet it does not guarantee optimal distillation quality at every stage. To investigate the influence of teacher models at different stages, we compute the trajectory loss gradients of the distilled data under various alignment scopes and visualize their cosine similarities.

As shown in Figure 4 (a), the gradient similarities across different alignment starting points remain low along the original trajectories, indicating high variance in the update directions of the distilled data, even in the later, more stable training stages. This instability hinders effective and efficient knowledge transfer, further corroborating the observations in Section 2.

To address this, we propose *Shortcut Trajectory*. Rather than fitting the full trajectory directly, it retains key endpoint information and introduces intermediate teacher models with stronger structure and guidance. These intermediates assist in aligning endpoints, resulting in new trajectories with more stable optimization targets and clearer update directions.

Formally, we set an endpoint $t_p$ for stage $p$. The trajectory $\tau_{\text{conv}}^p = \{\theta_t^p \mid 0 \le t \le t_p\}$ is defined as

$$\theta_t^p = (1 - \beta_p(t))\theta_0 + \beta_p(t)\theta_{t_p}, \quad (4)$$

where $\beta_p(t) \in (0,1)$ serves as a weighting function that governs the placement of waypoints along the trajectory. Here, the start and end points, $\theta_0$ and $\theta_{t_p}$, are exactly $\theta_0^p$ and $\theta_{t_p}^p$ from $\tau_{\text{conv}}$, respectively. In essence, $\tau_{\text{conv}}$ provides a direct interpolation from $\theta_0$ to $\theta_{t_p}$.

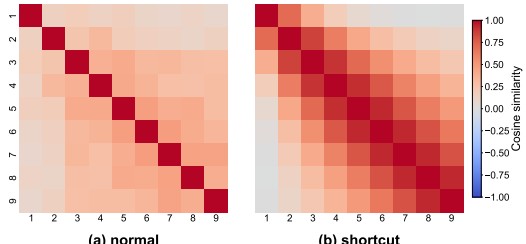

Figure 4: Gradient cosine similarity on the synthetic dataset across different epochs for the original and shortcut trajectories.

The weighting function $\beta_p(t)$ is computed proportionally based on the accumulated distance along the original trajectory $\tau_{\text{mtt}}$, as follows:

$$\beta_p(0) = 0, \quad \beta_p(t) = \frac{\sum_{l=0}^{t-1} \text{Norm}(\theta_{l+1} - \theta_l)}{\sum_{l=0}^{t_p-1} \text{Norm}(\theta_{l+1} - \theta_l)}, \quad (5)$$

where $\text{Norm}(\cdot)$ denotes the $\ell_2$-norm. To mitigate inconsistencies across different network layers, we compute layer-wise $\ell_2$-normalization, yielding a vector $\beta_p(t) = [\beta_p^1(t), \beta_p^2(t), \dots, \beta_p^L(t)]^\top$, where each component corresponds to a distinct layer parameter in the network.

Unlike Matching Convexified Trajectory (Zhong et al., 2025) interpolation strategy, which uses only the last point of the teacher's trajectory, we use different interpolation endpoints for each stage. As illustrated in Figure 3 (d), this strategy dynamically generates matching trajectories at each stage by utilizing the earliest and latest teacher models of the current phase, thereby effectively capturing inter-epoch variations in the teacher parameters.

As shown in Figure 4 (b)[1], the gradient similarity of the trajectory generated by our method is significantly improved, noting the optimization process of distillation dataset becomes more stable.

In addition, we have mathematically proven this gradient phenomenon, which can be summarised in the following proposition. The details and proof of this proposition are given in Appendix C.

**Proposition 1.** *Let $\ell(\tilde{\mathcal{D}}, \theta)$ denote the comparative learning loss on the distillation dataset $\tilde{\mathcal{D}}$ when the model parameter is $\theta$. Let $\mathcal{L}_1(\tilde{\mathcal{D}}), \mathcal{L}_2(\tilde{\mathcal{D}})$ denote the matching loss at two different ranges on the* **interpolated trajectory**, *with the difference between the starting points as $\Delta t$, $M$ is the maximum length of matching trajectory, then under the following assumptions:*

1. *The loss $\ell(\tilde{\mathcal{D}}, \theta)$ is twice continuously differentiable in $\theta$, and its Hessian $H_t := \nabla_{\theta\theta}^2 \ell(\tilde{\mathcal{D}}, \theta_t)$ along the trajectory $\{\theta_t\}$ satisfies a Lipschitz condition $\|H_{t'} - H_t\| \le L_H |t' - t|$, and in addition $\|H_t\| \le H_{\max}$ for some constant $H_{\max}$;*

2. *The mixed Hessian $\nabla_{\tilde{\mathcal{D}}\theta}^2 \ell(\tilde{\mathcal{D}}, \theta)$ has spectral norm bounded by $L_{\tilde{\mathcal{D}}\theta}$.*

*Then the following conclusion holds:*

$$\left\|\nabla_{\tilde{\mathcal{D}}}\mathcal{L}_2(\tilde{\mathcal{D}}) - \nabla_{\tilde{\mathcal{D}}}\mathcal{L}_1(\tilde{\mathcal{D}})\right\| \leq K\,\Delta t + \mathcal{O}(\Delta t^2), \text{ where } K = \frac{\eta}{M\,\|\Delta\theta\|}\big(L_H\|\Delta\theta\| + L_{\tilde{\mathcal{D}}\theta}\big). \quad (6)$$

This proposition states that the gradient difference of the interpolated trajectory converges linearly with $\Delta t$, but the original trajectory does not have this guarantee. This demonstrates that our method can effectively control gradient differences and enhance the stability of the distillation process.

To further improve the stability of the distillation process, we also used an exponential moving average smoothing strategy (Hunter, 1986). The distillation dataset $\tilde{\mathcal{D}}_p^i$ after the $i$-th iteration was smoothed and updated as follows.

$$\hat{\mathcal{D}}_p^i = \alpha\hat{\mathcal{D}}_p^{i-1} + (1-\alpha)\tilde{\mathcal{D}}_p^i \quad (7)$$

In summary, our procedure is shown in Algorithm 1.

## 4 EXPERIMENT

### 4.1 EEXPERIMENTAL SETTINGS

**Dataset and task.** We evaluate our method on two standard vision–language benchmarks: Flickr30K (Plummer et al., 2015) and COCO (Lin et al., 2014), using the Karpathy split (Karpathy & Fei-Fei, 2015) for training, validation, and testing. Flickr30k and COCO are image captioning datasets with 31K and 123K images respectively, and each image is paired with five captions. The model performance is commonly measured by the recall of top-K retrieval (R@K): given a query from one modality, we retrieve the closest k matches from the other modality and measure the correctness. We denote the text-to-image retrieval as IR@K and image-to-text retrieval as TR@K.

**Baseline method.** We compare our method against several baselines. For coreset selection, we include random subset selection (*Random*), Herd (Welling, 2009), K-center (Farahani & Hekmatfar, 2009), and the forgetting strategy (*Forgetting*) (Toneva et al., 2018). For dataset distillation, we consider MTT-VL Wu et al. (2023), LoRS Xu et al. (2024) and EDGM (Zhao et al., 2025). A detailed description of these methods can be found in the Appendix

**Implementation Details.** We evaluate data-pair counts $N \in \{100, 200, 500\}$ with an EMA decay of 0.99. All experiments run on a single 3090 GPU, highlighting the method's efficiency and low memory use. For details regarding time and memory overhead, please refer to Appendix D.6.

Following prior work (Xu et al., 2024), we mainly employ a trainable Normalizer-Free ResNet (NFNet) (Brock et al., 2021) as the image encoder with ImageNet-pretrained weights (Deng et al., 2009). For the text encoder, we use a pretrained and frozen BERT (Devlin et al., 2019), followed by a trainable projection head. We also tried other encoders such as RegNet (Radosavovic et al., 2020) and DistilBERT (Sanh et al., 2019) in Appendix D.5. For additional training details and more analysis, please refer to Appendix D.

### 4.2 MAIN RESULTS

The main results are summarized in Table 1, Table 2 and Table 3. Experimental results demonstrate that our PTM-ST method consistently yields significant gains across all metrics, as shown in our empirical evaluations below. Under equal compression ratios, the multi-stage distillation strategy outperforms conventional single-stage baselines, confirming the effectiveness and robustness of the phase-aware mechanism in MDD. This advantage grows with the number of synthetic samples, indicating that decomposing distillation into semantically aligned phases allows each subset to capture the teacher model's evolving dynamics at different training stages. Scaling experiments to 1,000 distilled pairs further validates the scalability and efficacy of our approach. Comparative analysis of core-set selection methods shows negligible or worse performance relative to random sampling, underscoring their limitations in modeling complex cross-modal dynamics, under our standard metrics

---

[1] We have provided the drawing details for Figure 2 and Figure 4 in Appendix B.2.

Table 1: Results on Flickr30k. The metrics for training model on the full dataset are IR@1=27.3, IR@5=57.1, IR@10=69.7; TR@1=33.9, TR@5=65.1, TR@10=75.2.

| Pairs | Ratio | Metric | Core-set Selection | | | | Dataset Distillation | | | | |
|---|---|---|---|---|---|---|---|---|---|---|---|
| | | | RAND | HERD | K-CENT | FORGET | MTT-VL[2] | LoRS[3] | EDGE[2] | PTM-ST | △ |
| 100 | 0.3% | IR@1 | 1.0 | 0.7 | 0.7 | 0.7 | $4.7_{\pm0.2}$ | $7.8_{\pm0.2}$ | - | $\mathbf{9.6_{\pm0.3}}$ | +1.8 |
| | | IR@5 | 4.0 | 2.8 | 3.1 | 2.4 | $15.7_{\pm0.5}$ | $24.4_{\pm0.3}$ | - | $\mathbf{28.4_{\pm0.5}}$ | +4.0 |
| | | IR@10 | 6.5 | 5.3 | 6.1 | 5.6 | $24.6_{\pm1.0}$ | $35.5_{\pm0.4}$ | - | $\mathbf{41.5_{\pm0.5}}$ | +6.0 |
| | | TR@1 | 1.3 | 1.1 | 0.6 | 1.2 | $9.9_{\pm0.3}$ | $10.2_{\pm0.3}$ | - | $\mathbf{14.4_{\pm0.4}}$ | +4.2 |
| | | TR@5 | 5.9 | 4.7 | 4.2 | 4.2 | $28.3_{\pm0.5}$ | $30.9_{\pm0.8}$ | - | $\mathbf{38.6_{\pm0.8}}$ | +7.7 |
| | | TR@10 | 10.1 | 7.9 | 7.6 | 9.7 | $39.1_{\pm0.7}$ | $44.9_{\pm0.6}$ | - | $\mathbf{52.7_{\pm0.7}}$ | +7.8 |
| 200 | 0.7% | IR@1 | 1.1 | 1.5 | 1.5 | 1.2 | $4.6_{\pm0.9}$ | $10.8_{\pm0.5}$ | - | $\mathbf{12.5_{\pm0.1}}$ | +1.7 |
| | | IR@5 | 4.8 | 5.5 | 5.4 | 3.1 | $16.0_{\pm1.6}$ | $29.8_{\pm0.4}$ | - | $\mathbf{34.7_{\pm0.2}}$ | +4.9 |
| | | IR@10 | 9.2 | 9.3 | 9.9 | 8.4 | $25.5_{\pm2.6}$ | $40.0_{\pm0.7}$ | - | $\mathbf{48.5_{\pm0.2}}$ | +8.5 |
| | | TR@1 | 2.1 | 2.3 | 2.2 | 1.5 | $10.2_{\pm0.8}$ | $14.1_{\pm0.5}$ | - | $\mathbf{19.3_{\pm0.2}}$ | +5.2 |
| | | TR@5 | 8.7 | 8.4 | 8.2 | 8.4 | $28.7_{\pm1.0}$ | $36.1_{\pm0.4}$ | - | $\mathbf{45.9_{\pm0.4}}$ | +9.8 |
| | | TR@10 | 13.2 | 14.4 | 13.5 | 10.2 | $41.9_{\pm1.9}$ | $50.0_{\pm0.3}$ | - | $\mathbf{58.9_{\pm0.5}}$ | +8.9 |
| 500 | 1.7% | IR@1 | 2.4 | 3.0 | 3.5 | 1.8 | $6.6_{\pm0.3}$ | $12.7_{\pm0.4}$ | 6.7 | $\mathbf{16.0_{\pm0.2}}$ | +3.3 |
| | | IR@5 | 10.5 | 10.0 | 10.4 | 9.0 | $20.2_{\pm1.2}$ | $32.9_{\pm0.3}$ | 21.0 | $\mathbf{40.5_{\pm0.3}}$ | +7.6 |
| | | IR@10 | 17.4 | 17.0 | 17.3 | 15.9 | $30.0_{\pm2.1}$ | $44.9_{\pm0.8}$ | 30.5 | $\mathbf{54.0_{\pm0.3}}$ | +9.1 |
| | | TR@1 | 5.2 | 5.1 | 4.6 | 3.6 | $13.3_{\pm0.6}$ | $14.7_{\pm0.6}$ | 13.3 | $\mathbf{22.2_{\pm0.6}}$ | +7.5 |
| | | TR@5 | 18.3 | 16.4 | 16.4 | 12.3 | $32.8_{\pm1.8}$ | $37.6_{\pm0.7}$ | 35.6 | $\mathbf{51.1_{\pm0.3}}$ | +14.0 |
| | | TR@10 | 25.7 | 24.3 | 23.3 | 19.3 | $46.8_{\pm0.8}$ | $51.1_{\pm0.4}$ | 47.5 | $\mathbf{64.6_{\pm0.3}}$ | +13.5 |

Table 2: Results on the COCO dataset. The metrics for training model on the full dataset are IR@1=16.9, IR@5=41.9, IR@10=55.9; TR@1=19.6, TR@5=45.6, TR@10=59.5.

| Pairs | Ratio | Metric | Core-set Selection | | | | Dataset Distillation | | | | |
|---|---|---|---|---|---|---|---|---|---|---|---|
| | | | RAND | HERD | K-CENT | FORGET | MTT-VL[2] | LoRS[3] | EDGE[2] | PTM-ST | △ |
| 100 | 0.8‰ | IR@1 | 0.3 | 0.5 | 0.4 | 0.3 | $1.3_{\pm0.1}$ | $1.7_{\pm0.1}$ | - | $\mathbf{2.3_{\pm0.1}}$ | +0.6 |
| | | IR@5 | 1.3 | 1.4 | 1.4 | 1.5 | $5.4_{\pm0.3}$ | $6.9_{\pm0.2}$ | - | $\mathbf{9.0_{\pm0.1}}$ | +2.1 |
| | | IR@10 | 2.7 | 3.5 | 2.5 | 2.5 | $9.5_{\pm0.5}$ | $11.9_{\pm0.3}$ | - | $\mathbf{15.6_{\pm0.3}}$ | +3.7 |
| | | TR@1 | 0.8 | 0.8 | 1.4 | 0.7 | $2.5_{\pm0.3}$ | $3.0_{\pm0.3}$ | - | $\mathbf{4.1_{\pm0.2}}$ | +1.1 |
| | | TR@5 | 3.0 | 2.1 | 3.7 | 2.6 | $10.0_{\pm0.5}$ | $11.0_{\pm0.2}$ | - | $\mathbf{13.4_{\pm0.2}}$ | +2.4 |
| | | TR@10 | 5.0 | 4.9 | 5.5 | 4.8 | $15.7_{\pm0.4}$ | $18.5_{\pm0.3}$ | - | $\mathbf{22.0_{\pm0.2}}$ | +3.5 |
| 200 | 1.7‰ | IR@1 | 0.6 | 0.9 | 0.7 | 0.6 | $1.7_{\pm0.1}$ | $2.2_{\pm0.2}$ | - | $\mathbf{3.9_{\pm0.1}}$ | +1.7 |
| | | IR@5 | 2.3 | 2.4 | 2.1 | 2.8 | $6.5_{\pm0.4}$ | $8.6_{\pm0.1}$ | - | $\mathbf{13.7_{\pm0.2}}$ | +5.1 |
| | | IR@10 | 4.4 | 4.1 | 5.8 | 4.9 | $12.3_{\pm0.8}$ | $14.7_{\pm0.2}$ | - | $\mathbf{22.2_{\pm0.2}}$ | +7.5 |
| | | TR@1 | 1.0 | 1.0 | 1.2 | 1.1 | $3.3_{\pm0.2}$ | $3.6_{\pm0.3}$ | - | $\mathbf{5.7_{\pm0.2}}$ | +2.1 |
| | | TR@5 | 4.0 | 3.6 | 3.8 | 3.5 | $11.9_{\pm0.6}$ | $12.1_{\pm0.2}$ | - | $\mathbf{18.2_{\pm0.3}}$ | +6.1 |
| | | TR@10 | 7.2 | 7.7 | 7.5 | 7.0 | $19.4_{\pm1.2}$ | $20.8_{\pm0.4}$ | - | $\mathbf{27.8_{\pm0.3}}$ | +7.0 |
| 500 | 4.4‰ | IR@1 | 1.1 | 1.7 | 1.1 | 0.8 | $2.5_{\pm0.5}$ | $3.3_{\pm0.2}$ | 1.8 | $\mathbf{6.6_{\pm0.1}}$ | +3.3 |
| | | IR@5 | 5.0 | 5.3 | 6.3 | 5.8 | $8.9_{\pm0.7}$ | $11.8_{\pm0.4}$ | 6.5 | $\mathbf{20.5_{\pm0.2}}$ | +8.7 |
| | | IR@10 | 8.7 | 9.9 | 10.5 | 8.2 | $15.8_{\pm1.5}$ | $19.2_{\pm0.6}$ | 11.2 | $\mathbf{30.7_{\pm0.2}}$ | +11.5 |
| | | TR@1 | 3.2 | 3.8 | 3.7 | 2.3 | $5.0_{\pm0.4}$ | $4.1_{\pm0.5}$ | 2.9 | $\mathbf{6.9_{\pm0.3}}$ | +1.9 |
| | | TR@5 | 7.5 | 7.8 | 8.7 | 8.2 | $17.2_{\pm1.3}$ | $12.8_{\pm0.4}$ | 9.5 | $\mathbf{20.1_{\pm0.2}}$ | +2.9 |
| | | TR@10 | 12.5 | 13.7 | 13.8 | 13.0 | $26.0_{\pm1.9}$ | $20.2_{\pm0.9}$ | 15.7 | $\mathbf{30.0_{\pm0.3}}$ | +4.0 |

and settings. In contrast, our progressive framework explicitly transfers multi-phase knowledge from early and late teacher stages, significantly enhancing the distilled dataset's representational capacity. Notably, on Flickr30K, PTM-ST achieves 76% of full training performance using only 1.7% of the original data, highlighting its potential to improve compression efficiency in multimodal distillation. In addition, we extend our method to the larger dataset LLaVA-cc3m (Liu et al., 2023). This dataset is prepared for LLaVA pre-training and contains 595k image-text pairs. We split it into training, validation, and test sets in a 3:1:1 ratio. We also explore more powerful DiNo-v2 (Oquab et al., 2023) as the image encoder and BGE-1.5 (Xiao et al., 2024) as the text encoder in Appendix D.4. The results shown in Table 3 and Table 16 reveal that our method remains effective with scaled training data size and model capacity, and it consistently outperforms the previous SOTA.

## 4.3 ABLATION STUDY

This section presents an ablation study on the three core components of our framework: Progressive Teacher Modeling (PTM), Shortcut Trajectory (ST), and Exponential Moving Average (EMA). Experiments conducted on 500 distilled pairs from Flickr30K and COCO are summarized in Table 4. The results demonstrate that each component contributes positively to the overall performance, validating their individual effectiveness in enhancing distillation quality.

---

[2]Intercepted from the original paper.
[3]Reproduced by ourselves.

Table 3: Results on LLaVA-cc3m dataset. The metrics for training model on the full dataset are IR@1=9.3, IR@5=25.9, IR@10=36.5; TR@1=9.8, TR@5=26.4, TR@10=37.3.

| Pairs | LoRS | | | | | | Ours | | | | | |
|---|---|---|---|---|---|---|---|---|---|---|---|---|
| | IR@1 | IR@5 | IR@10 | TR@1 | TR@5 | TR@10 | IR@1 | IR@5 | IR@10 | TR@1 | TR@5 | TR@10 |
| 100 | 1.2 | 4.6 | 7.7 | 1.7 | 6.9 | 11.4 | 2.3 | 8.2 | 13.2 | 2.9 | 10.0 | 15.9 |
| 200 | 1.4 | 5.3 | 8.7 | 2.4 | 8.5 | 13.6 | 2.7 | 9.7 | 15.8 | 3.7 | 11.9 | 18.4 |
| 500 | 1.7 | 6.2 | 10.1 | 2.5 | 8.7 | 13.8 | 3.3 | 11.4 | 17.9 | 4.1 | 13.2 | 19.9 |

Table 4: Various ablation studies with 500 pairs on Flickr30k and COCO.

| No. | Model | Flickr IR@K | | | Flickr TR@K | | | COCO IR@K | | | COCO TR@K | | |
|---|---|---|---|---|---|---|---|---|---|---|---|---|---|
| | | 1 | 5 | 10 | 1 | 5 | 10 | 1 | 5 | 10 | 1 | 5 | 10 |
| (1) | BASE (N/A) | 12.2 | 33.0 | 45.7 | 16.2 | 39.4 | 54.0 | 3.4 | 11.7 | 19.1 | 4.2 | 13.8 | 20.8 |
| (2) | EMA | 12.9 | 33.7 | 46.3 | 16.2 | 40.6 | 54.3 | 3.5 | 12.3 | 19.8 | 4.0 | 13.6 | 20.8 |
| (3) | PTM | 13.4 | 35.2 | 48.1 | 19.6 | 43.2 | 55.5 | 4.2 | 14.2 | 22.6 | 4.8 | 15.1 | 23.4 |
| (4) | ST | 14.2 | 37.8 | 50.8 | 19.5 | 45.1 | 59.3 | 4.9 | 15.9 | 24.7 | 4.8 | 15.1 | 23.8 |
| (5) | PTM + EMA | 14.3 | 37.7 | 50.7 | 18.8 | 45.0 | 58.5 | 4.5 | 14.8 | 23.1 | 4.8 | 15.7 | 24.3 |
| (6) | ST + EMA | 15.4 | 38.3 | 51.4 | 19.7 | 46.7 | 60.3 | 5.2 | 17.0 | 26.3 | 5.0 | 15.8 | 24.0 |
| (7) | PTM + ST | 15.4 | 38.8 | 52.2 | 22.3 | 50.5 | 64.6 | 6.3 | 20.1 | 30.2 | 6.2 | 19.9 | 29.8 |
| (8) | OURS(PTM + ST + EMA) | **15.5** | **39.6** | **53.6** | **22.9** | **51.6** | **64.9** | **6.6** | **20.5** | **30.7** | **6.9** | **20.1** | **30.0** |

Among the modules, Shortcut Trajectory (ST) provides the largest improvement by stabilizing and guiding the teacher model's optimization path, which is crucial for reliable and representative knowledge transfer. PTM further boosts performance by dividing the teacher's training into phases to better align semantic information across stages. EMA helps smooth training dynamics and improves robustness in multimodal settings. Together, these components work synergistically to achieve the best distillation results. Quantitatively, Table 4 shows monotonic gains from EMA to PTM to ST, with PTM+ST contributing most of the improvement. Adding EMA on top yields small but consistent benefits, giving the best IR@K/TR@K on both datasets.

### 4.4 GENERALIZATION TO VQA AND ZERO-SHOT CLASSIFICATION

To more comprehensively evaluate the generalization capability of our distillation method, we further conducted experiments on Visual Question Answering (COCO-QA) (Lu et al., 2016) and ImageNet classification (Deng et al., 2009). Specifically, in COCO-QA, we enumerate all possible answers and construct the template "Question: {question} Answer: {answer}." for each question–answer pair, and then compute the Top-K retrieval performance. For ImageNet, we randomly sample 50 categories from ImageNet-1K and report the Top-K zero-shot classification accuracies.

In these comparisons, teacher denotes the model trained on the full COCO dataset and evaluated directly in a zero-shot manner on both tasks. In contrast, LoRS and Ours are trained under the same distillation budget, using only 499 COCO image–text pairs. As summarized in Table 5, under identical supervision budgets, our method significantly outperforms LoRS and further narrows the performance gap relative to the full-data teacher model.

### 4.5 DISTILLATION FOR MORE PAIRS

To further validate the scalability of our approach, we extend the distillation process to 1,000 synthetic data points distributed across three distinct training phases (details in Appendix D). Table 6 presents a comparative analysis between our proposed method and the baseline LoRS. While conventional methods such as LoRS tend to saturate or even experience performance degradation as the number of distilled samples increases—evidenced by the declining IR@K scores—our phased

Table 5: Performance on additional downstream tasks.

| Methods | VQA on COCO-QA | | | ImageNet-50 Classification | | |
|---|---|---|---|---|---|---|
| | ACC@1 | ACC@5 | ACC@10 | ACC@1 | ACC@5 | ACC@10 |
| LoRS | 10.8 | 33.0 | 39.9 | 18.6 | 42.1 | 55.9 |
| Ours | **16.3** | **38.9** | **47.4** | **22.1** | **52.7** | **67.9** |

Table 6: Performance comparison on Flickr30k and COCO with 1000 pairs.

| Dataset | Method | IR@1 | IR@5 | IR@10 | TR@1 | TR@5 | TR@10 |
|---------|--------|------|------|-------|------|------|-------|
| Flickr30k | LoRS | 9.8 | 29.0 | 41.6 | 14.9 | 39.8 | 53.5 |
| | EDGE | 9.9 | 28.2 | 40.5 | 14.5 | 38.3 | 51.7 |
| | Ours | **17.5** | **42.8** | **56.3** | **23.8** | **53.6** | **67.2** |
| COCO | LoRS | 2.5 | 9.9 | 16.6 | 4.2 | 15.2 | 24.1 |
| | EDGE | 2.8 | 9.8 | 16.2 | 3.9 | 13.0 | 21.0 |
| | Ours | **7.0** | **21.8** | **32.3** | **6.8** | **20.9** | **30.8** |

distillation strategy consistently yields performance improvements. This demonstrates the effectiveness of our approach in capturing the evolving knowledge of the teacher model across different training stages, thereby enhancing the quality of the distilled dataset even at larger scales.

## 4.6 VISUALIZATION

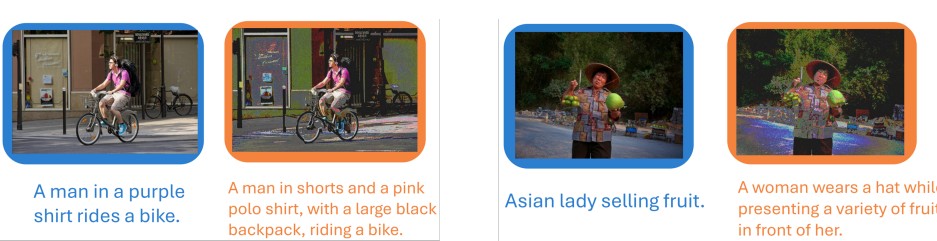

A man in a purple shirt rides a bike. | A man in shorts and a pink polo shirt, with a large black backpack, riding a bike. | Asian lady selling fruit. | A woman wears a hat while presenting a variety of fruits in front of her.

Figure 5: Examples of initial (left) and synthetic (right) image-text pairs.

We visualize the images and texts of the synthetic pairs from Flickr30k to illustrate the distilled data. Figure 5 presents the images and texts before (initial) and after distillation. The distilled images exhibit a DeepDream-like style (Zeiler & Fergus, 2014), a phenomenon commonly observed in dataset distillation, often accompanied by repetitive textures, exaggerated details, and hallucinatory patterns. The distilled texts, in contrast, are richer and more informative. Other examples are in Appendix F.

## 5 CONCLUSION

In this study, we propose PTM-ST, a phased distillation framework that addresses a key limitation in multimodal dataset distillation (MDD): the inability of conventional methods to capture and transfer the complex, evolving knowledge from teacher models during later training stages. It introduces Phased Teacher Modeling for stage-aware guidance and a Shortcut Trajectory for stabilizing teacher evolution. Experiments on Flickr30K and MS-COCO show that PTM-ST outperforms existing methods in both performance and robustness.

**Future Work.** PTM-ST requires manual specification of interpolation endpoints and matching ranges for each stage, which may result in excessive parameterisation and difficulty in tuning. Future research could focus on how to adaptively adjust these parameters as the distillation process progresses, transforming it into an automated, continuous procedure. Furthermore, to provide new perspectives, future work could explore distillation methods not based on MTT (see Appendix E).

## ACKNOWLEDGEMENT

This work was supported by the Guangdong Basic and Applied Basic Research Foundation (2025A1515011546) and by the Shenzhen Science and Technology Program (JCYJ20240813105901003, KJZD20240903102901003, ZDCY20250901113000001).

## REPRODUCIBILITY STATEMENT

We describe the methodology we applied in detail in Section 3, and provide parameter information in Appendix D. The dataset Flickr30k was downloaded from here, while COCO was obtained from here. Our code is primarily built upon the foundations laid by previous work (Xu et al., 2024; Cazenavette et al., 2022). We are currently organising the code and will make it publicly available in the near future.

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

# A   RELATED WORK DETAILS

**Multimodal learning.**   Multimodal Learning (ML) has demonstrated superior representational power and holds substantial promise for a wide range of applications (Jia et al., 2021; Chen et al., 2023c; Li et al., 2025; Lai et al., 2024b), owing to its ability to integrate complementary information from diverse modalities. Recent works have further extended this capability to reasoning segmentation (Lai et al., 2024a; Yang et al., 2023) and open-vocabulary perception (Wang et al., 2025a;b). Existing multimodal learning approaches predominantly seek to project inputs into a unified feature space, thereby facilitating the alignment of matched samples while distinguishing unmatched ones (Shao et al., 2024b; Cui et al., 2023a). However, compared to unimodal learning tasks such as semantic segmentation and object detection (Tian et al., 2020; 2022b; Cui et al., 2022; Peng et al., 2023), multimodal data is not only larger in scale and richer in information, but also significantly more complex (Lei et al., 2024). This increased complexity often compromises data quality and continuity, thereby exacerbating the challenges associated with training, such as object hallucinations (Peng et al., 2025). Consequently, there is an urgent need to explore more efficient learning strategies (Yang et al., 2025a; Tian et al., 2022a; Yang et al., 2025b) that are better tailored to the unique characteristics of multimodal scenarios (Zolfaghari et al., 2021; Lin & Hu, 2022; Yang et al., 2024).

**Image-Text Contrastive Learning.**   In multimodal data set distillation (Wu et al., 2023), the basic task is image-text contrast learning. Let the image-text pair dataset be $\mathcal{D} = (\mathcal{X}, \mathcal{Y})$, which contains $m$ pairs of images $\mathcal{X} = \{x_i\}_m$ and text $\mathcal{Y} = \{y_i\}_m$. Image-Text Contrastive models (e.g. CLIP) include an image encoder $f_V$ and a text encoder $f_T$, which encode image and text into one-dimensional vector representations, respectively, i.e. $u_i = f_V(x_i)$, $v_i = f_T(y_i)$. The model performs cross-modal retrieval using cosine similarity $\hat{s}_{ij} = \cos(u_i, v_j)$. During training, the InfoNCE loss (Oord et al., 2018) is typically used:

$$L_{\text{NCE}}(\mathcal{B}) = -\frac{1}{b} \sum_{i=1}^{b} \log(P_{ii}^V) + \log(P_{ii}^T), \quad P_{ij}^V = \frac{\exp(\hat{s}_{ij}/\tau_c)}{\sum_k \exp(\hat{s}_{ik}/\tau_c)}, \quad P_{ij}^T = \frac{\exp(\hat{s}_{ij}/\tau_c)}{\sum_k \exp(\hat{s}_{kj}/\tau_c)}, \tag{8}$$

where $\mathcal{B} \subset \mathcal{D}$ is a small batch of data with a size of $b$. $P_{ij}^V$ and $P_{ij}^T$ are the softmax probabilities of similarity $\hat{s}_{ij}$, respectively, and $\tau_c$ is the temperature factor. InfoNCE assumes that the paired text $y_i$ for each image $x_i$ is a positive sample, while other texts $y_k, k \neq i$ are negative samples, thereby achieving contrastive alignment.

**Match Training Trajectory.**   The MTT (Cazenavette et al., 2022) method guides the optimization of the synthetic dataset $\tilde{\mathcal{D}}$ by training two separate models on $\tilde{\mathcal{D}}$ and the real dataset $\mathcal{D}$, respectively, and matching their weight trajectories of different lengths—namely, a trajectory of length $t$ on $\tilde{\mathcal{D}}$ and a trajectory of length $\Delta T$ on $\mathcal{D}$.

**Low-Rank Similarity Mining.**   The recent method LoRS (Xu et al., 2024) further improves the expressive power of the distillation dataset by introducing a learnable similarity matrix. And use wBCE loss when training student models:

$$L_{\text{wBCE}}(\tilde{\mathcal{B}}, \tilde{S}) = \frac{1}{|\{s_{ij} > \beta\}|} \sum_{i,j:s_{ij}>\beta} \ell\big(s_{ij}, \sigma(\hat{s}_{ij}/\tau)\big) + \frac{1}{|\{s_{ij} \leq \beta\}|} \sum_{i,j:s_{ij}\leq\beta} \ell\big(s_{ij}, \sigma(\hat{s}_{ij}/\tau)\big),$$

where $\tilde{\mathcal{B}} \subset \tilde{\mathcal{D}}$ is a small batch of data in synthetic dataset, $\beta$ is the positive/negative threshold, set to 0.5, $\sigma$ is the sigmoid function, $s_{ij}$ is the learnable similarity between the $i$-th image and the $j$-th text in $\tilde{\mathcal{D}}$, and $\ell(y, p) = -y \log(p) - (1 - y) \log(1 - p)$.

They perform low-rank decomposition on the similarity matrix, i.e., $\tilde{S} = \omega I + \frac{a}{r} L R^\top$, where $w, L_{N \times r}, R_{N \times r}$ are learnable parameters, and $a, r$ are hyperparameters. The number of parameters saved by using low-rank decomposition is very small, and our method only uses the complete similarity matrix. As shown in Table 12 (sim_type=full).

**Efficient Multimodal Dataset Distillation via GEnerative Models.** Zhao et al. (2025) proposes a generative multimodal dataset distillation framework called EDGE, which efficiently replaces the original large-scale graphic dataset with a very small number of generated samples while maintaining the retrieval performance by training a graphic generation model and introducing a bi-directional contrast loss, diversity loss and caption synthesis strategy.

Although this form of supervision primarily aims to capture cross-modal relationships, it inadvertently introduces additional information such as intra-modal similarities. The inconsistency of these signals between the early and late stages of training significantly increases the training difficulty. Moreover, existing methods like LoRS only capture early training dynamics, synthesizing samples using weights from the first two epochs. While the teacher model continues to improve with further training, the student model rapidly collapses during the distillation process.

## B  MOTIVATION

Section B.1 primarily introduces the motivation behind our entire project and the process of exploring methods. Section B.2 introduces the specific settings of the visualisation experiment.

### B.1  MOTIVATION DETAILS

Our overall approach stems from an important observation: LoRS only utilises the teacher model from the first 20–30% of the training phase during the distillation of synthetic datasets. This differs significantly from traditional single-modal dataset distillation methods, which typically rely on teacher information from the entire training process. However, when we attempted to directly introduce the teacher model from the later training stages into the distillation process, we observed severe instability or even collapse in the training process of the student model, as shown in Table 7. This phenomenon prompted us to delve deeper into the question: why does the introduction of the later teacher model lead to such severe performance degradation in multimodal tasks?

Table 7: Performance of teacher and student models at different epochs on COCO and Flickr30K.

| Dataset | Model | 2 | 4 | 6 | 8 | 10 |
|---|---|---|---|---|---|---|
| COCO | Teacher | 35.4 | 39.0 | 40.4 | 41.9 | 42.8 |
| | Student | 11.4 | 7.9 | 6.9 | 5.7 | 5.1 |
| Flickr30K | Teacher | 45.6 | 48.9 | 51.2 | 53.6 | 55.3 |
| | Student | 33.8 | 35.1 | 33.5 | 31.4 | 30.1 |

**DATM Experiment.** We referenced the dynamic alignment strategy proposed by DATM (Guo et al., 2023) in single-modal data distillation: as distillation proceeds, dynamically increase the sampling upper bound $T^+$ of the matching starting point. This gradually captures the training dynamics of the teacher at different training stages. We directly applied DATM to existing multimodal data set distillation frameworks, and the experimental results on COCO are shown in Table 8. Simply expanding the sampling range of matching starting points through the distillation process still reduces the distillation effect.

**More Visualization.** To better understand the root causes of these challenges, we conducted a detailed analysis by visualizing the loss gradients that were back-propagated from the teacher model to the synthetic dataset across multiple training epochs, as shown in Figure 6. Our observations revealed that, as training progressed, the gradient magnitudes steadily increased. More critically, the direction of these gradients oscillated violently between epochs, exhibiting no discernible pattern. This erratic behavior suggests that the training dynamics of the teacher model are inherently unstable, and such instability is directly inherited by the synthetic dataset.

**Remark.** The above analysis inspires us: due to the complexity of multimodal data, the information in different training stages of the teacher model has significant differences. It is necessary to develop a distillation method that can capture the training dynamics of different stages.

Table 8: Results on the COCO dataset. The metrics for training the model on the full dataset are IR@1=16.9, IR@5=41.9, IR@10=55.9; TR@1=19.6, TR@5=45.6, TR@10=59.5.

| Pairs | Ratio | Metric | Dataset Distillation | | | |
|---|---|---|---|---|---|---|
| | | | MTT-VL[1] | LoRS[2] | DATM[2] | PTM-ST |
| 100 | 0.8‰ | IR@1 | 1.3±0.1 | 1.7±0.1 | 1.2±0.1 | **2.3±0.1** |
| | | IR@5 | 5.4±0.3 | 6.9±0.2 | 5.1±0.2 | **9.0±0.1** |
| | | IR@10 | 9.5±0.5 | 11.9±0.3 | 8.9±0.4 | **15.6±0.3** |
| | | TR@1 | 2.5±0.3 | 3.0±0.3 | 2.5±0.3 | **4.1±0.2** |
| | | TR@5 | 10.0±0.5 | 11.0±0.2 | 9.1±0.3 | **13.4±0.2** |
| | | TR@10 | 15.7±0.4 | 18.5±0.3 | 14.9±0.3 | **22.0±0.2** |
| 200 | 1.7‰ | IR@1 | 1.7±0.1 | 2.2±0.2 | 1.6±0.2 | **3.9±0.1** |
| | | IR@5 | 6.5±0.4 | 8.6±0.1 | 6.3±0.2 | **13.7±0.2** |
| | | IR@10 | 12.3±0.8 | 14.7±0.2 | 11.8±0.2 | **22.2±0.2** |
| | | TR@1 | 3.3±0.2 | 3.6±0.3 | 2.8±0.2 | **5.7±0.2** |
| | | TR@5 | 11.9±0.6 | 12.1±0.2 | 11.1±0.4 | **18.2±0.3** |
| | | TR@10 | 19.4±1.2 | 20.8±0.4 | 17.8±0.3 | **27.8±0.3** |
| 500 | 4.4‰ | IR@1 | 2.5±0.5 | 3.3±0.2 | 2.2±0.3 | **6.6±0.1** |
| | | IR@5 | 8.9±0.7 | 11.8±0.4 | 8.1±0.4 | **20.5±0.2** |
| | | IR@10 | 15.8±1.5 | 19.2±0.6 | 14.3±0.7 | **30.7±0.2** |
| | | TR@1 | 5.0±0.4 | 4.1±0.5 | 3.2±0.4 | **6.9±0.3** |
| | | TR@5 | 17.2±1.3 | 12.8±0.4 | 10.9±0.6 | **20.1±0.2** |
| | | TR@10 | 26.0±1.9 | 20.2±0.9 | 18.3±1.0 | **30.0±0.3** |

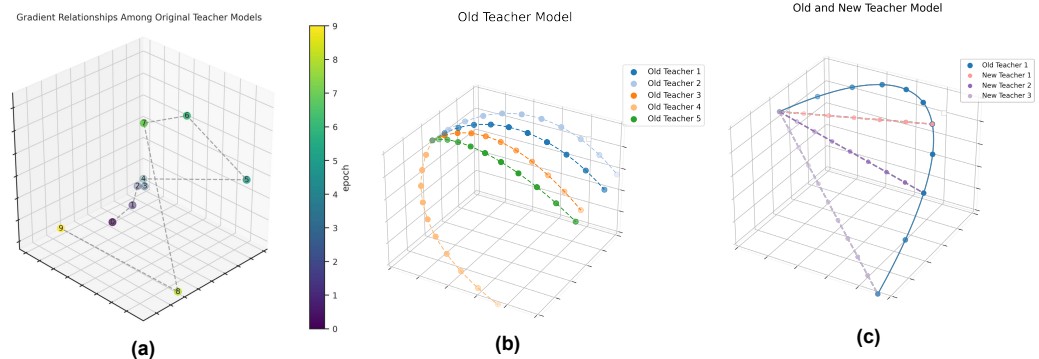

Figure 6: **(a)** shows the gradients of the teacher model on the synthetic dataset across different epochs. **(b)** illustrates the parameter differences between various training trajectories of the teacher model. **(c)** present a visualization of the new teacher model constructed using our proposed method.

## B.2 EXPLAIN OF FIGURE

**Explain of 3D.** As illustrated in Figure 6, **(a)** presents a detailed visualization of the gradients propagated from the teacher model to the synthetic dataset at various training epochs. It is apparent that as the training process progresses, the magnitude of the gradients increases substantially. More importantly, the directions of these gradients exhibit erratic fluctuations, with no discernible pattern, highlighting the inherent instability in the teacher model's learning trajectory as it advances through later training epochs. This behavior further reinforces our earlier assertion that the teacher model becomes increasingly volatile during the later stages of training, a phenomenon that significantly complicates the distillation process for the student model.

In **(b)**, we depict the parameter differences between teacher models that have been derived from distinct training trajectories. The substantial divergence observed between these trajectories serves as an additional confirmation of the inconsistencies and instability inherent in independently trained teacher models. This variability across trajectories further emphasizes the challenges associated

with applying standard distillation techniques, as it introduces significant misalignment between the knowledge encapsulated in different teacher models.

**(c)** provides a visualization of the parameter trajectory of the new teacher model generated using our proposed method. The resulting shortcut trajectory, constructed with the strategic interpolation of key teacher checkpoints, exhibits remarkable stability when compared to traditional teacher trajectories. This enhanced stability directly contributes to the consistency of the distilled knowledge, enabling the student model to receive more reliable and coherent supervision signals throughout the training process. Consequently, this improvement in trajectory consistency facilitates a more effective knowledge transfer, ultimately leading to better performance and robust distillation results.

**Figure in Main Paper.** Figure 2 (a) in the paper illustrates the performance differences between the teacher model and the student model on the Flickr30k test set. The specific values are shown in Table 7. The teacher model was trained directly on the real dataset, with the x-axis coordinates representing the number of epochs trained. The student model was trained on the distilled dataset, where the epoch count on the x-axis represents the maximum value of the trajectory matching starting point when generating this distilled dataset (max_start_epoch / $T^+$). PTM-ST was not used when generating this distilled dataset. The remaining parameters are shown in Table 9.

Table 9: Hyper-parameters for the distillation run.

| Parameter | Flickr30k | COCO |
|---|---|---|
| syn_steps | 8 | 8 |
| expert_epochs | 1 | 1 |
| lr_img | 1000 | 1000 |
| lr_txt | 1000 | 1000 |
| lr_lr | 1e-2 | 1e-2 |
| lr_teacher_img | 0.1 | 0.1 |
| lr_teacher_txt | 0.1 | 0.1 |
| lr_sim | 10 | 50 |
| sim_type | lowrank | lowrank |
| sim_rank | 20 | 40 |
| alpha | 0.01 | 1.0 |
| num_queries | 499 | 499 |
| mini_batch_size | 40 | 30 |
| loss_type | wBCE | wBCE |

In Figure 2 (b), we used the original expert trajectories to calculate the gradient of the trajectory matching loss on the distilled dataset under different matching ranges, and visualised it using PCA dimensionality reduction. Specifically, we selected the first 200 images from Flickr30K as the initialisation for the synthetic dataset, and calculated the trajectory matching loss $\mathcal{L}_{MTT}$ using different trajectory matching starting points (as indicated by the numbers in the figure). We then backpropagated this loss to the distilled dataset to obtain the gradient.

In Figure 4, we calculate the gradient on the synthetic dataset using the original expert trajectory and the trajectory after interpolation. The interpolation endpoint is the endpoint of the original trajectory. We extract the gradient of the image part and calculate the cosine similarity between each pair, and plot the result in Figure 4 in the paper.

## C  PROOF OF THE PROPOSITION

**Notation.** Let the original teacher trajectory be $\tau = \{\theta_0, \theta_1, \ldots, \theta_n\}$. We reparameterize this trajectory by linear interpolation:

$$\theta_t = \theta_0 + t\,\Delta\theta, \quad t \in [0, n], \quad \Delta\theta := \theta_n - \theta_0.$$

Thus the direction is constant at all times (i.e. a straight line with uniform speed).

For a given distilled dataset $\tilde{\mathcal{D}}$, one SGD step with step size $\eta$ is denoted:

$$\tilde{\theta}_t(\tilde{\mathcal{D}}) = \theta_t - \eta\,g_t, \quad g_t := \nabla_\theta \ell(\tilde{\mathcal{D}}, \theta_t),$$

where $\ell$ is the training loss (e.g. InfoNCE or wBCE).

---

[1]Intercepted from the original paper.

[2]Reproduced by ourselves.

**Trajectory-Matching Loss.** We wish to match a trajectory segment of length $M$ so that the student move approximates the "target step" $M\Delta\theta$. For two starting points $t$ and $t + \Delta t$, the matching losses are

$$\mathcal{L}_1(\tilde{\mathcal{D}}) = \frac{\left\|\tilde{\theta}_t - \theta_t - M_1\,\Delta\theta\right\|}{M_1\,\|\Delta\theta\|}, \tag{9}$$

$$\mathcal{L}_2(\tilde{\mathcal{D}}) = \frac{\left\|\tilde{\theta}_{t+\Delta t} - \theta_{t+\Delta t} - M_2\,\Delta\theta\right\|}{M_2\,\|\Delta\theta\|}. \tag{10}$$

Substituting the SGD update gives:

$$v_1 := \eta\,g_t + M_1\,\Delta\theta, \quad v_2 := \eta\,g_{t+\Delta t} + M_2\,\Delta\theta, \tag{11}$$

so that:

$$\mathcal{L}_1(\tilde{\mathcal{D}}) = \frac{\|v_1\|}{M_1\|\Delta\theta\|}, \quad \mathcal{L}_2(\tilde{\mathcal{D}}) = \frac{\|v_2\|}{M_2\|\Delta\theta\|}.$$

**Gradients w.r.t. the distilled dataset $\tilde{\mathcal{D}}$.** By differentiating the norm and applying the chain rule, one obtains:

$$\nabla_{\tilde{\mathcal{D}}}\mathcal{L}_1(\tilde{\mathcal{D}}) = \frac{1}{M_1\|\Delta\theta\|}\frac{v_1^\top}{\|v_1\|}\left(-\eta\,\nabla_{\tilde{\mathcal{D}}}g_t\right), \tag{12}$$

$$\nabla_{\tilde{\mathcal{D}}}\mathcal{L}_2(\tilde{\mathcal{D}}) = \frac{1}{M_2\|\Delta\theta\|}\frac{v_2^\top}{\|v_2\|}\left(-\eta\,\nabla_{\tilde{\mathcal{D}}}g_{t+\Delta t}\right). \tag{13}$$

**Main Proposition: Linear convergence of gradient-difference on the interpolated trajectory.**

**Proposition 2.** *Under the following assumptions:*

1. *The loss $\ell(\tilde{\mathcal{D}}, \theta)$ is twice continuously differentiable in $\theta$, and its Hessian*

$$H_t := \nabla^2_{\theta\theta}\,\ell(\tilde{\mathcal{D}}, \theta_t)$$

   *along the trajectory $\{\theta_t\}$ satisfies a Lipschitz condition $\|H_{t'} - H_t\| \leq L_H\,|t' - t|$, and in addition $\|H_t\| \leq H_{\max}$ for some constant $H_{\max}$;*

2. *The mixed Hessian $\nabla^2_{\tilde{\mathcal{D}}\theta}\,\ell(\tilde{\mathcal{D}}, \theta)$ has spectral norm bounded by $L_{\tilde{\mathcal{D}}\theta}$.*

*Then for the interpolated trajectory $\theta_t = \theta_0 + t\Delta\theta$ ($\Delta\theta = \theta_n - \theta_0$), one has*

$$\left\|\nabla_{\tilde{\mathcal{D}}}\mathcal{L}_2(\tilde{\mathcal{D}}) - \nabla_{\tilde{\mathcal{D}}}\mathcal{L}_1(\tilde{\mathcal{D}})\right\| \leq K\,\Delta t + \mathcal{O}(\Delta t^2), \quad K = \frac{\eta}{M\,\|\Delta\theta\|}\left(L_H\,\|\Delta\theta\| + L_{\tilde{\mathcal{D}}\theta}\right). \tag{14}$$

*Proof.* For brevity set

$$g_t := \nabla_\theta\ell(\tilde{\mathcal{D}}, \theta_t), \quad G_t := \nabla_{\tilde{\mathcal{D}}}g_t = \nabla^2_{\tilde{\mathcal{D}}\theta}\,\ell(\tilde{\mathcal{D}}, \theta_t), \quad H_t := \nabla^2_{\theta\theta}\,\ell(\tilde{\mathcal{D}}, \theta_t).$$

We bound $\|\nabla_{\tilde{\mathcal{D}}}\mathcal{L}_2 - \nabla_{\tilde{\mathcal{D}}}\mathcal{L}_1\|$ in three steps.

**(i) Taylor Expansion.** Along the interpolation path $\theta_{t+\Delta t} = \theta_t + \Delta t\,\Delta\theta$, we write $g(\theta) = \nabla_\theta\ell(S, \theta)$ and expand to second order:

$$g_{t+\Delta t} = g(\theta_t + \Delta t\,\Delta\theta) = g(\theta_t) + \int_0^1 H\left(\theta_t + s\,\Delta t\,\Delta\theta\right)\left(s\,\Delta t\,\Delta\theta\right)\mathrm{d}s,$$

where $H(\theta) = \nabla^2_{\theta\theta}\ell(S, \theta)$. Split the integral at $H(\theta_t)$:

$$g_{t+\Delta t} = g_t + H_t\left(\Delta t\,\Delta\theta\right) + \underbrace{\int_0^1\left[H(\theta_t + s\,\Delta t\,\Delta\theta) - H_t\right]\left(s\,\Delta t\,\Delta\theta\right)\mathrm{d}s}_{R_t}.$$

Since $H$ is $L_H$-Lipschitz along the path, $\big\| H(\theta_t + s\,\Delta t\,\Delta\theta) - H_t \big\| \leq L_H\,(s\,\Delta t)\,\|\Delta\theta\|$, we bound the remainder:

$$\|R_t\| \leq \int_0^1 L_H\,(s\,\Delta t)\,\|\Delta\theta\| \times \big\| s\,\Delta t\,\Delta\theta \big\|\,\mathrm{d}s = L_H\,\|\Delta\theta\|^2(\Delta t)^2 \int_0^1 s^2\,\mathrm{d}s = \frac{L_H}{3}\,\|\Delta\theta\|^2\,(\Delta t)^2 \tag{15}$$

So that:

$$g_{t+\Delta t} = g_t + \Delta t\,H_t\,\Delta\theta + \tfrac{1}{2}(\Delta t)^2\,R_t, \quad \|R_t\| \leq L_H\,\|\Delta\theta\|. \tag{16}$$

Since $\|H_t\| \leq H_{\max}$, the term $\Delta t\,H_t\,\Delta\theta$ is itself $\mathcal{O}(\Delta t)$.

**(ii) Gradient expressions.** Using the definition $\tilde{\theta}_i = \theta_i - \eta g_i$ and the chain rule,

$$\mathcal{L}_i(\tilde{\mathcal{D}}) = \frac{\|\eta g_i + M_i \Delta\theta\|}{M_i \|\Delta\theta\|}, \quad \nabla_{\tilde{\mathcal{D}}}\mathcal{L}_i = -\frac{\eta}{M_i \|\Delta\theta\|}\,\hat{v}_i^\top\,G_i,$$

where $v_i = \eta g_i + M_i \Delta\theta$ and $\hat{v}_i = v_i/\|v_i\|$. Hence

$$\nabla_{\tilde{\mathcal{D}}}\mathcal{L}_2 - \nabla_{\tilde{\mathcal{D}}}\mathcal{L}_1 = -\frac{\eta}{\|\Delta\theta\|}\Big( \tfrac{1}{M_2}\hat{v}_2^\top G_{t+\Delta t} - \tfrac{1}{M_1}\hat{v}_1^\top G_t \Big).$$

**(iii) Bounding the difference.** From *equation* 16 and normalization one shows $\hat{v}_2 = \hat{v}_1 + \mathcal{O}(\Delta t)$, and similarly

$$G_{t+\Delta t} = G_t + \Delta t\,\dot{G}_t + \mathcal{O}(\Delta t^2), \quad \|\dot{G}_t\| \leq L_{\tilde{\mathcal{D}}\theta}\,\|\Delta\theta\|.$$

Applying the triangle inequality and Cauchy–Schwarz yields

$$\|\nabla_{\tilde{\mathcal{D}}}\mathcal{L}_2 - \nabla_{\tilde{\mathcal{D}}}\mathcal{L}_1\| \leq \frac{\eta}{M\|\Delta\theta\|}\Big( \|\hat{v}_1\|\|\dot{G}_t\| + \|G_t\|\|\hat{v}_2 - \hat{v}_1\| \Big)\Delta t + \mathcal{O}(\Delta t^2).$$

Here $M = \min\{M_1, M_2\}$, so that the denominator $M\|\Delta\theta\|$ scales the bound by the size of that segment. Since $\|\hat{v}_1\| = 1$ and $\|G_t\| \leq L_{\tilde{\mathcal{D}}\theta}$, setting

$$K = \frac{\eta}{M\,\|\Delta\theta\|}\big( L_H\,\|\Delta\theta\| + L_{\tilde{\mathcal{D}}\theta} \big)$$

completes the proof of *equation* 14. $\qquad\square$

**Remark.** On the *non-interpolated* trajectory, each segment direction $\Delta\theta_p$ may differ substantially from the global $\Delta\theta$. This effectively amplifies $\|\hat{v}_2 - \hat{v}_1\|$ by a factor $\alpha > 1$, leading to a larger constant in the bound and thus explaining the empirically observed lower gradient cosine-similarity.

## D  EXPERIMENT

For LoRS and PTM-ST, similarity matrices require additional memory. To ensure fairness, we remove one data pair, saving 151K parameters $(3 \times 224^2 + 768)$. In PTM-ST, no phase split is used for 100 pairs; we use $(99 + 100)$ for 200 and $(200 + 299)$ for 500. For 500 pairs, the similarity matrix size is 129K, smaller than one sample's footprint. This ensures a memory-fair comparison across all settings.

### D.1  HYPER-PARAMETERS

This section shows the specific settings we obtained for the teacher trajectory and distillation process hyper-parameters, as shown in Table 10 and Table 12. The ema_decay is 0.99 for all experiment. We use the SGD optimizer to train the model with momentum=0.9, weight_decay=0.0005. For the student training, we adopt hyper-parameters shown in Table 11. It should be noted that during the distillation process, the learning rate of the training student model is learnable, and this learning rate is lr_lr=0.01, and the table shows the values taken at the time of initialization.

Table 10: Hyperparameters for teacher training.

| | Flickr-30K | MS-COCO |
|---|---|---|
| epoch / $n$ | 10 | 10 |
| num_experts | 20 | 20 |
| batch_size | 128 | 128 |
| lr_teacher_img | 0.1 | 0.1 |
| lr_teacher_txt | 0.1 | 0.1 |
| image_size | 224×224 | 224×224 |

Table 11: Hyperparameter for student training.

| | Flickr-30K | MS-COCO |
|---|---|---|
| epoch (per subset) | 50 | 50 |
| batch_size | 128 | 128 |
| lr_img | 0.1 | 0.1 |
| lr_txt | 0.1 | 0.1 |
| image_size | 224×224 | 224×224 |

Table 12: Hyper-parameter settings for distillation.

| | Flickr-30K | | | | MS-COCO | | | |
|---|---|---|---|---|---|---|---|---|
| | 100 | 200 | 500 | 1000 | 100 | 200 | 500 | 1000 |
| subset_num / $P$ | 1 | 2 | 2 | 3 | 1 | 2 | 2 | 3 |
| syn_steps / $t$ | 8 | 8 | 8 | 8 | 8 | 8 | 8 | 8 |
| expert_epochs / $\Delta T$ | 1 | 1 | 1 | 1 | 1 | 1 | 1 | 1 |
| min_start_epoch / $T_p^-$ | 0 | 0, 1 | 0, 1 | 0, 1, 2 | 0 | 0, 1 | 2 | 0, 1, 2 |
| max_start_epoch / $T_p^+$ | 2 | 2, 3 | 2, 3 | 3, 4, 5 | 2 | 2, 3 | 2 | 2, 3, 4 |
| iteration / $I_p$ | 2000 | 2000*2 | 2000*2 | 2000*3 | 2000 | 2000*2 | 2000*2 | 2000,500,1000 |
| interpolation endpoints / $t_p$ | 6 | 6, 8 | 6, 8 | 6, 8, 10 | 6 | 6, 8 | 6, 8 | 6, 8, 10 |
| lr_img | 1000 | 1000 | 1000 | 1000 | 1000 | 1000 | 1000 | 1000 |
| lr_txt | 1000 | 1000 | 1000 | 1000 | 1000 | 1000 | 5000 | 1000 |
| lr_lr | 1e-2 | 1e-2 | 1e-2 | 1e-2 | 1e-2 | 1e-2 | 1e-2 | 1e-2 |
| lr_teacher_img | 0.1 | 0.1 | 0.1 | 0.1 | 0.1 | 0.1 | 0.1 | 0.1 |
| lr_teacher_txt | 0.1 | 0.1 | 0.1 | 0.1 | 0.1 | 0.1 | 0.1 | 0.1 |
| lr_sim | 10.0 | 10.0 | 10.0 | 10.0 | 50.0 | 50.0 | 50.0 | 50.0 |
| sim_type | full | full | full | full | full | full | full | full |
| num_queries / $N_p$ | 99 | 99,100 | 200,299 | 332,332,333 | 99 | 99,100 | 200,299 | 200,299,499 |
| mini_batch_size | 20 | 20 | 40 | 40 | 20 | 20 | 20 | 20 |
| loss_type | wBCE | wBCE | wBCE | wBCE | wBCE | wBCE | wBCE | wBCE |

## D.2 ADDITIONAL ABLATION STUDY

In our method, the distillation process is divided into stages, with each stage having different sampling ranges and interpolation endpoints for trajectory matching. As shown in Table 12, the $T_p^-$, $T_p^+$, and $t_p$ values for each stage are shifted backward. To demonstrate this, we conducted additional ablation experiments on Flickr30K, keeping all other parameters constant while omitting this shift, and compared the distillation results. The results are shown in Table 13, where PTM without shift indicates that $T_p^-$ and $T_p^+$ are the same for each stage, and ST without shift indicates that $t_p$ is the same for each stage.

Table 13: Ablation on whether to shift under different data sizes.

| Pairs | Setting | IR@1 | IR@5 | IR@10 | TR@1 | TR@5 | TR@10 | Mean |
|---|---|---|---|---|---|---|---|---|
| 200 | PTM w/o shift | 11.8 | 33.2 | 46.3 | 18.1 | 44.3 | 59.0 | 35.5 |
| | ST w/o shift | 11.6 | 33.3 | 47.0 | 18.5 | 43.5 | 57.8 | 35.3 |
| | full | 12.5 | 34.7 | 48.5 | 19.3 | 45.9 | 58.9 | **36.6** |
| 500 | PTM w/o shift | 15.2 | 39.6 | 53.1 | 21.6 | 48.6 | 62.9 | 40.2 |
| | ST w/o shift | 15.3 | 39.9 | 53.5 | 21.1 | 49.6 | 63.1 | 40.4 |
| | full | 16.0 | 40.5 | 54.0 | 22.2 | 51.1 | 64.6 | **41.4** |
| 1000 | PTM w/o shift | 16.9 | 41.9 | 55.4 | 21.6 | 49.3 | 62.5 | 41.3 |
| | ST w/o shift | 16.8 | 42.7 | 56.5 | 23.6 | 51.4 | 64.9 | 42.7 |
| | full | 17.5 | 42.8 | 56.3 | 23.8 | 53.6 | 67.2 | **43.5** |

As can be seen from the table, both the sampling range and the offset of the interpolation endpoint can improve the distillation effect. This confirms that our method is conducive to capturing the dynamics of the model's later training in the synthetic dataset.

## D.3 COMPARISON OF DIFFERENT INTERPOLATION ENDPOINTS AND MATCHING RANGES

**Interpolation endpoints.** Our approach necessitates specifying interpolation endpoints for each stage in advance. Consequently, we established several sets of distinct interpolation endpoint parameters for each pair of distilled models and conducted experiments on the Flickr-30k dataset. All

other parameters remained identical to those in Table 12, with the experimental results presented in Table 14.

Table 14: Comparison of distillation results at different interpolation endpoints.

| Pairs | Interpolation endpoint | IR@1 | IR@5 | IR@10 | TR@1 | TR@5 | TR@10 | Mean |
|---|---|---|---|---|---|---|---|---|
| 100 | 6 | 9.6 | 28.4 | 41.5 | 14.4 | 38.6 | 52.7 | **30.9** |
| | 8 | 8.4 | 26.6 | 39.7 | 15.4 | 39.9 | 54.6 | 30.8 |
| | 10 | 8.8 | 26.8 | 39.7 | 14.9 | 39.6 | 53.6 | 30.6 |
| 200 | 4, 6 | 11.7 | 33.2 | 46.3 | 18.5 | 43.5 | 58.6 | 35.3 |
| | 6, 8 | 12.5 | 34.7 | 48.5 | 19.3 | 45.9 | 58.9 | **36.6** |
| | 8, 10 | 11.2 | 32.3 | 45.4 | 18.1 | 46.4 | 60.2 | 35.6 |
| 500 | 4, 6 | 15.3 | 39.6 | 53.1 | 22.6 | 49.2 | 63.4 | 40.5 |
| | 6, 8 | 16.0 | 40.5 | 54.0 | 22.2 | 51.1 | 64.6 | **41.4** |
| | 8, 10 | 14.9 | 38.8 | 52.2 | 22.1 | 51.2 | 64.4 | 40.6 |
| 1000 | 4, 7, 10 | 16.6 | 42.5 | 56.2 | 23.7 | 50.2 | 62.5 | 42.0 |
| | 6, 8, 10 | 17.5 | 42.8 | 56.3 | 23.8 | 53.6 | 67.2 | **43.5** |
| | 8, 9, 10 | 17.1 | 43.2 | 57.1 | 23.2 | 52.8 | 65.7 | 43.2 |

The result demonstrates that the selection of interpolation endpoints also exerts a certain influence on the final results. It is necessary to select appropriate values to ensure the distilled dataset can capture the training information corresponding to the teacher model's respective stages.

**Matching ranges.** For the matching range $(T_p^-, T_p^+)$, we adopt a simple linear strategy in our experiments: $T_p^- = T_1^- + \Delta d(p - 1)$, $T_p^+ = T_1^+ + \Delta d(p - 1)$. To further assess the effect of varying $\Delta d$, we conduct additional ablation studies under the Flickr30k 500-pair setting and the results are shown in Table 15.

Table 15: Effect of $\Delta d$ on retrieval performance.

| Method | IR@1 | IR@5 | IR@10 | TR@1 | TR@5 | TR@10 | Mean |
|---|---|---|---|---|---|---|---|
| $\Delta d = 0$ | 15.2 | 39.6 | 53.1 | 21.6 | 48.6 | 62.9 | 40.2 |
| $\Delta d = 1$ | 16.0 | 40.5 | 54.0 | 22.2 | 51.1 | 64.6 | 41.4 |
| $\Delta d = 2$ | 15.9 | 40.1 | 53.2 | 21.0 | 50.1 | 63.7 | 40.7 |
| LoRS | 12.7 | 32.9 | 44.9 | 14.7 | 37.6 | 51.1 | 32.3 |

The results reveal that although different parameter choices introduce some variation, the overall influence is limited; more importantly, our method consistently outperforms the LoRS baseline across all configurations. This demonstrates the robustness and practicality of our approach.

## D.4 DISTILATION ON POWERFUL VISION-LANGUAGE MODELS

To further validate the applicability of our method to larger-scale models, we conduct additional experiments using the more powerful DINO-v2 (Oquab et al., 2023) and BGE-1.5 (Xiao et al., 2024) as the image and text encoders, respectively. Due to computing resource constraints, both encoders are kept frozen, and a single-layer trainable projection head was added on top of them.

Table 16: Results on DiNo-v2 + BGE-1.5 on MS-COCO. The model trained on the full dataset performs: IR@1=22.5, IR@5=50.8, IR@10=65.0, TR@1=31.7, TR@5=61.4, TR@10=74.0.

| Pairs | LoRS | | | | | | Ours | | | | | |
|---|---|---|---|---|---|---|---|---|---|---|---|---|
| | IR@1 | IR@5 | IR@10 | TR@1 | TR@5 | TR@10 | IR@1 | IR@5 | IR@10 | TR@1 | TR@5 | TR@10 |
| 100 | 5.8 | 20.6 | 30.7 | 10.2 | 29.4 | 39.8 | 8.1 | 23.5 | 34.9 | 13.5 | 33.8 | 46.4 |
| 200 | 6.1 | 20.9 | 32.1 | 12.0 | 31.4 | 43.1 | 8.4 | 24.1 | 35.4 | 14.1 | 35.6 | 48.4 |
| 500 | 7.7 | 22.9 | 33.3 | 12.7 | 33.5 | 46.1 | 9.1 | 25.9 | 37.2 | 15.0 | 36.5 | 50.1 |

As shown in Table 16, evaluation on the COCO dataset shows that our method remains effective even when scaling up the model capacity. This demonstrates that the proposed approach generalizes well across different backbone sizes.

## D.5 Cross Architecture Generalization

To validate the generalisation performance of the synthetic dataset, we conducted cross-model evaluation. The data is distilled with NFNet+BERT and evaluated on other architectures such as RegNet (Radosavovic et al., 2020) and DistilBERT (Sanh et al., 2019). As shown in Table 17, our method demonstrates good generalisation performance and outperforms the baseline method.

Table 17: Cross architecture generalization.

| Method | Model | IR@1 | IR@5 | IR@10 | TR@1 | TR@5 | TR@10 | Mean |
|--------|-------|------|------|-------|------|------|-------|------|
| LoRS | NFNet + BERT | 12.1 | 33.2 | 45.9 | 16.6 | 42.2 | 55.7 | 34.3 |
| | NFNet + DistilBERT | 7.6 | 24.5 | 34.7 | 12.7 | 34.5 | 46.1 | 26.7 |
| | RegNet + BERT | 4.5 | 15.0 | 23.7 | 5.4 | 21.0 | 32.4 | 17.0 |
| Ours | NFNet + BERT | 14.6 | 39.1 | 52.5 | 21.5 | 50.6 | 64.0 | **40.4** |
| | NFNet + DistilBERT | 11.3 | 29.8 | 41.3 | 19.0 | 41.3 | 56.5 | **33.2** |
| | RegNet + BERT | 4.2 | 14.4 | 22.3 | 8.7 | 23.8 | 36.0 | **18.2** |

## D.6 Time and Memory Overhead

We recorded the computational time and memory consumption of our proposed method and the LoRS method across key experiments, as shown in Table 18. Specifically, we documented the average time required per iteration to update the synthetic dataset during distillation (in seconds per iteration) and the peak and average memory consumption throughout the process (in gigabytes). It can be observed that our method incurs slightly lower time and VRAM costs than the original approach. This is attributable to our phased division, wherein each phase distils only a subset of data, eliminating the need to store computationally intensive information across an entire phase.

Table 18: Runtime and memory comparison across different numbers of pairs.

| Method | time (s/iter) | | | peak memory (GB) | | | avg memory (GB) | | |
|--------|------|------|------|--------|--------|--------|-------|-------|-------|
| | 100 | 200 | 500 | 100 | 200 | 500 | 100 | 200 | 500 |
| LoRS | 4.522 | 4.913 | 7.392 | 13.713 | 16.333 | 20.942 | 5.339 | 5.586 | 7.985 |
| Ours | 4.570 | 4.554 | 7.250 | 13.768 | 13.818 | 20.363 | 5.390 | 5.404 | 7.592 |

# E An Attempt at the NCFM Method

**Introduction.** Neural Characteristic Function Matching (NCFM) (Wang et al., 2025c) has recently demonstrated outstanding performance in uni-modal dataset distillation. It first uses an encoder to extract high-dimensional vector representations of images, then calculates the Neural Characteristic Function Discrepancy (NCFD) as the matching loss between the real and synthetic datasets. It employs a minmax adversarial update strategy to optimise the frequency sampling network to maximise this loss, while updating the synthetic dataset to minimise it. The distillation process can be represented as follows:

$$
\min_{\tilde{\mathcal{D}}} \max_{\psi} \mathcal{L}(\tilde{\mathcal{D}}, \mathcal{D}, f, \psi) = \min_{\tilde{\mathcal{D}}} \max_{\psi} \mathbb{E}_{x \sim \mathcal{D}, \tilde{x} \sim \tilde{\mathcal{D}}} \mathcal{C}_{\mathcal{T}}(x, \tilde{x}; f, \psi)
$$
$$
= \min_{\tilde{\mathcal{D}}} \max_{\psi} \mathbb{E}_{x \sim \mathcal{D}, \tilde{x} \sim \tilde{\mathcal{D}}} \int_t \sqrt{\mathrm{Chf}(t; f)} \, dF_{\mathcal{T}}(t; \psi), \tag{17}
$$

where $\mathrm{Chf}(t) = (\Phi_x(t) - \Phi_{\tilde{x}}(t))(\bar{\Phi}_x(t) - \bar{\Phi}_{\tilde{x}}(t))$, and can be reformulated as:

$$
\mathrm{Chf}(t; f) = \alpha \left( \left|\left| \Phi_{f(x)}(t) - \Phi_{f(\tilde{x})}(t) \right|\right|^2 \right)
$$
$$
+ (1 - \alpha) \cdot \left( 2 \left| \Phi_{f(x)}(t) \right| \left| \Phi_{f(\tilde{x})}(t) \right| \right) \cdot \left( 1 - \cos \left( a_{f(x)}(t) - a_{f(\tilde{x})}(t) \right) \right). \tag{18}
$$

Here: $\Phi_x(t) = \mathbb{E}_x \left[ e^{j \langle t, x \rangle} \right] = \int_x e^{j \langle t, x \rangle} \, dF(x)$ is the characteristic function of $x$; $t$ is the frequency argument; $\left( 1 - \cos \left( a_{f(x)}(t) - a_{f(\tilde{x})}(t) \right) \right)$ denotes the phase difference; and $f$ is the image encoder.

**Our Attempts.** Uni-modal data set distillation is mainly used for classification tasks, where NCFM distils $ipc$ synthetic data for each category. Multi-modal data set distillation is used for

image-text matching retrieval tasks, which do not have category concepts, so it is not possible to directly apply this method.

To migrate it to multimodal data set distillation, we first perform clustering on the original image-text paired data set. Specifically, for each data point $(x_i, y_i)$, we use the trained teacher models $f_V^*$ and $f_T^*$ to extract its high-dimensional representations $u_i = f_V^*(x_i)$ and $v_i = f_T^*(y_i)$, and concatenate these two modalities' high-dimensional representations to obtain the vector representation of this data point. Then, we perform Mini-Batch K-Means clustering using this vector representation. During distillation, we treat each cluster as a category and extract $ipc$ data points from it for initialisation. We match feature functions of the data points using the same method as NCFM. After passing the encoder, the concatenation of the two modal vector representations corresponds to $x$ in Eq. 17.

**Results.**  We tested this method on Flickr30K. The learning rate for the sampling net and synthetic dataset was set to 0.1, the sampling frequency was set to 512, and the weight $\alpha$ of the phase difference term in the loss function was set to 0.5. The results are shown in Table 19.

Table 19: Results of the NCFM method on Flickr30K.

| Clusters | ipc | IR@1 | IR@5 | IR@10 | TR@1 | TR@5 | TR@10 | Mean |
|---|---|---|---|---|---|---|---|---|
| 100 | 1 | 0.7 | 3.2 | 5.7 | 2.1 | 7.1 | 10.5 | 4.9 |
| 100 | 2 | 1.4 | 5.5 | 9.6 | 2.9 | 9.3 | 14.5 | 7.2 |
| 100 | 5 | 3.0 | 10.7 | 17.9 | 4.8 | 15.8 | 25.8 | 13.0 |
| 200 | 1 | 1.4 | 5.5 | 9.6 | 2.9 | 9.3 | 14.5 | 7.2 |
| 200 | 2 | 2.5 | 9.6 | 15.6 | 5.6 | 15.9 | 23.8 | 12.2 |
| 500 | 1 | 3.2 | 11.7 | 18.9 | 6.1 | 18.1 | 26.0 | 14.0 |

The effectiveness of NCFM is inferior to that of methods based on trajectory matching. We speculate that this may be due to the sparsity of multimodal datasets and the more difficult retrieval tasks.

## F  VISUALIZATION OF DISTILLED DATA

In this section, we present visualizations of a subset of the synthetic dataset distilled by our proposed method. As illustrated in Figure 7 and Figure 8, the synthesized samples exhibit two notable characteristics. First, the textual descriptions become significantly more informative and semantically rich, providing more nuanced guidance for downstream learning. Second, the visual content contains subtle and complex perturbations that are difficult for the human eye to interpret, yet potentially essential for enhancing student model learning.

These findings suggest that our method does not simply replicate the teacher's outputs, but rather generates enriched supervision signals that span both linguistic and visual modalities. This further demonstrates the effectiveness of our approach in constructing high-quality synthetic data for multimodal knowledge distillation.

## G  LLM USAGE

We primarily utilise large language models to assist with writing tasks. Specific applications include:

- Inquiry regarding how to adjust the layout, such as spacing between elements like formulas, headings, algorithms, etc.
- Auxiliary table generation: Copy data from an Excel spreadsheet to an LLM to generate LaTeX table source code.
- To avoid excessive blank lines at the end of each line, inquire how the LLM might add or remove existing text.
- For some formulas, we employ large language models to generate preliminary versions, which are then manually revised.
- Enquire about LaTeX syntax, such as importing relevant packages and setting colours.

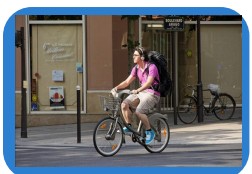 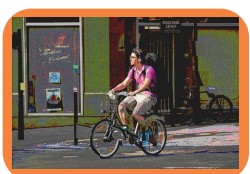 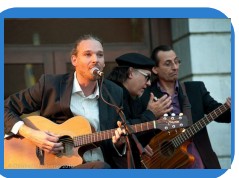 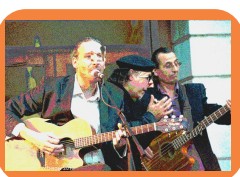

A man in a purple shirt rides a bike.

A man in shorts and a pink polo shirt, with a large black backpack and headphones, riding a bike.

A band sings and play their guitar.

A male musicians are performing with one playing a guitar and singing into a microphone, another holding a harmonica.

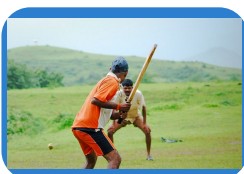 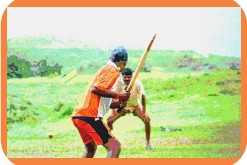 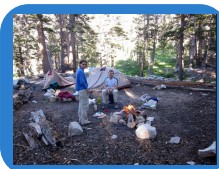 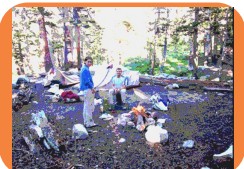

Two men playing a game of baseball.

In a rural village in India, two men are energetically playing a stick-ball game that resembles a casual form of baseball.

Two people sitting around a campfire.

Two people, one standing and one sitting, look at the camera, as they are situated around a campsite fire.

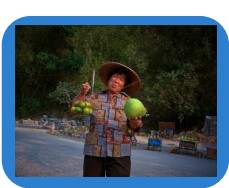 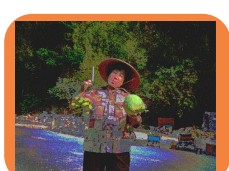 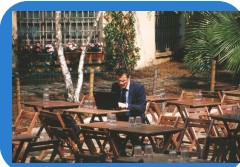 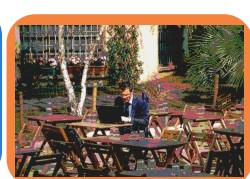

Asian lady selling fruit.

A woman wears a hat while presenting a variety of fruits in front of her, engaging with potential customers.

Man using a laptop at a table in an empty outdoor dining area.

A man in a suit and tie sits alone at a table, possibly working on his laptop in the serene setting.

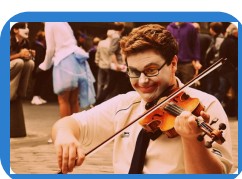 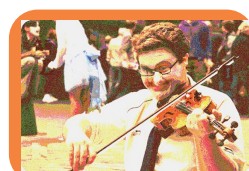 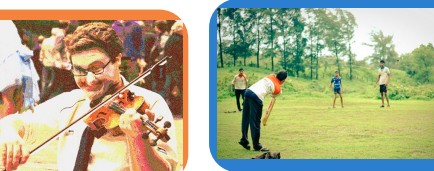 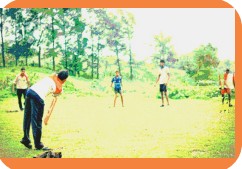

A Violinist plays as he is decorated as a clown.

A man dressed in formal attire with glasses and a tie passionately plays the violin.

Five young men in a grassy area playing a game.

A group of boys or young men play a field game together in a grassy open space.

Figure 7: Flickr-30K before and after distillation. (Left) The original image-text pairs before the distillation. (Right) The image-text pairs after distillation.

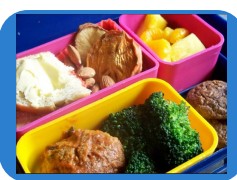
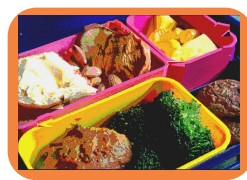
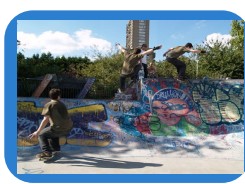
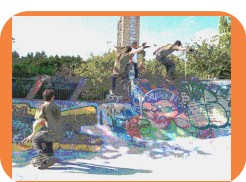

A meal is presented in brightly colored plastic trays.

A colorful assortment of food is presented in close-up inside bright plastic trays.

A young man riding a skateboard into the air.

A young man soars into the air on his skateboard, while nearby a group of teenagers also take turns jumping ramp.

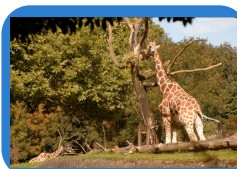
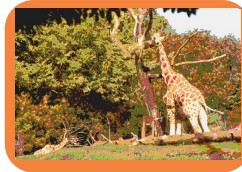
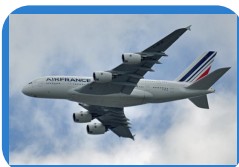
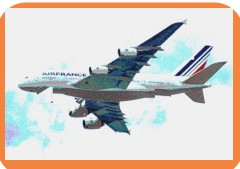

A giraffe standing up nearby a tree

A tall giraffe stretches its neck to feed from the top branches of a tree, standing gracefully in its natural habitat.

A big airplane flying in the big blue sky

A large airplane with four engines soars through the vast blue sky, showcasing its size and flight capabilities.

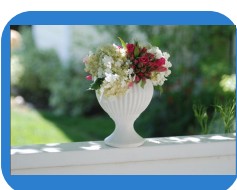
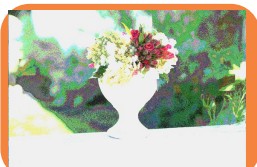
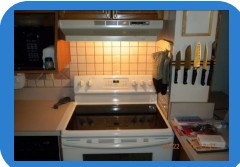
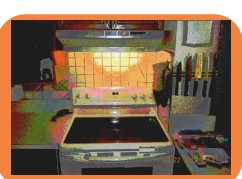

A flower vase is sitting on a porch stand.

A decorative white vase filled with various colored flowers is neatly placed on a stand out on a porch.

An oven with a stove on top of it in a kitchen.

A kitchen scene shows a stove with an oven below and a lighted hood above.

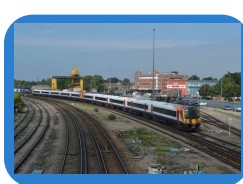
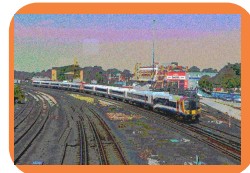
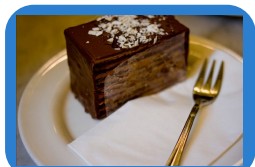
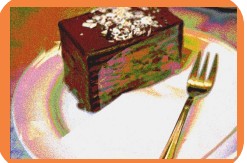

A train coming to a stop on the tracks out side.

A long train is approaching a stop on outdoor tracks, and it slows down on the railway line.

A chocolate cake and a fork ready to be eat

A delicious-looking piece of chocolate cake sits atop a clean white plate, accompanied by a fork.

Figure 8: COCO before and after distillation. (Left) The original image-text pairs before the distillation. (Right) The image-text pairs after distillation.

