# OpenReview forum: "Multimodal Dataset Distillation via Phased Teacher Models"
_ICLR.cc/2026/Conference — ICLR 2026 Poster_

### Official Review · Reviewer_SUGG · 2025-10-28

**Soundness:** 3
**Presentation:** 3
**Contribution:** 3
**Rating:** 4
**Confidence:** 3

**Summary:**

This paper proposes PTM-ST, which aims to address the instability of knowledge transfer from the teacher model during different training phases in MDD. The core idea lies in combining PTM and ST to capture the evolving knowledge of the teacher model across different training stages and to smooth its optimization trajectory, thereby enhancing the stability and effectiveness of knowledge distillation.

**Strengths:**

The paper conducts the first systematic analysis of the phenomenon of “phase-wise knowledge drift” in multimodal distillation, and validates its existence through both theoretical derivation and empirical evidence. The proposed method is evaluated on two mainstream datasets, Flickr30K and COCO, with consistent and significant performance gains demonstrated through comprehensive ablation studies.

**Weaknesses:**

1. The PTM-ST framework appears to be highly sensitive to the choices of phase partition ( P ), matching intervals, and endpoint selection. If these hyperparameters are not properly tuned, will the model’s performance collapse or degrade significantly? Can these parameters be made adaptive rather than manually specified?
2. Your Proposition 1 relies on the second-order smoothness assumption, which is often invalid for large Transformer architectures. How do you justify the practical relevance of this assumption, and have you conducted any empirical validation or relaxation of it?
3. The proposed method still requires access to the entire teacher training trajectory, which seems computationally expensive. If only partial teacher checkpoints or commercial API-based teacher models are available, can your method still be applied effectively, or does it fundamentally depend on the full training process?

**Questions:**

See wekness.

---

> ### Author Response · Authors · 2025-11-19
> **Response 1 - Q1**
>
> Thank you for your time and insightful feedback. We have provided detailed responses to all your questions below and hope they effectively resolve your concerns.
>
> **Q1 (Weaknesses 1): PTM-ST seems very sensitive to phase partitions, matching ranges, and endpoint choices. If not well tuned, performance may drop, raising the question of whether these hyperparameters can be made adaptive instead of manually set.**
>
> **A1:**
> Thank you for raising this point. In summary, our method is **not sensitive** to these hyperparameters, although we currently do not have a mechanism for adaptive tuning. To address concerns regarding parameter sensitivity, we conduct ablation studies on the Flickr-30k dataset.
>
> For the matching range, we adopt a simple linear strategy: $T_p^-=T_1^-+\Delta d(p-1)$, $T_p^+=T_1^++\Delta d(p-1)$, where $T_1^-$ and $T_1^+$ follow the settings of prior work, and we primarily focus on varying $\Delta d$. For the interpolation endpoint, we compare different choices in Appendix D.3.
>
> The results show that although varying these parameters introduces some fluctuations, the overall effect is limited; importantly, our method consistently outperforms the LoRS baseline. This demonstrates the **robustness and practical usability** of our approach. We will continue to explore the possibility of adaptive parameter selection in future work.
>
> We have ensured that these ablation results are included in Appendix D to maintain the completeness of the manuscript.
>
> - *Comparison of different matching ranges for 500 pairs, P=2.*
>
> |Method|IR@1|IR@5|IR@10|TR@1|TR@5|TR@10|mean|
> |-|-|-|-|-|-|-|-|
> |$\Delta d=0$|15.2|39.6|53.1|21.6|48.6|62.9|40.2|
> |$\Delta d=1$|16.0|40.5|54.0|22.2|51.1|64.6|41.4|
> |$\Delta d=2$|15.9|40.1|53.2|21.0|50.1|63.7|40.7|
> |LoRS|12.7|32.9|44.9|14.7|37.6|51.1|32.3|
>
> - *Appendix D.3: Comparison of different interpolation endpoints, $\Delta d=1$.*
>
> |Pairs|Interpolation Endpoint|IR@1|IR@5|IR@10|TR@1|TR@5|TR@10|Mean|LoRS Mean|
> |-|-|-|-|-|-|-|-|-|-|
> |100|6|9.6|28.4|41.5|14.4|38.6|52.7|30.9|25.6|
> ||8|8.4|26.6|39.7|15.4|39.9|54.6|30.8||
> ||10|8.8|26.8|39.7|14.9|39.6|53.6|30.6||
> |200|4, 6|11.7|33.2|46.3|18.5|43.5|58.6|35.3|30.1|
> ||6, 8|12.5|34.7|48.5|19.3|45.9|58.9|36.6||
> ||8, 10|11.2|32.3|45.4|18.1|46.4|60.2|35.6||
> |500|4, 6|15.3|39.6|53.1|22.6|49.2|63.4|40.5|32.3|
> ||6, 8|16.0|40.5|54.0|22.2|51.1|64.6|41.4||
> ||8, 10|14.9|38.8|52.2|22.1|51.2|64.4|40.6||
> |1000|4, 7, 10|16.6|42.5|56.2|23.7|50.2|62.5|42.0|31.4|
> ||6, 8, 10|17.5|42.8|56.3|23.8|53.6|67.2|43.5||
> ||8, 9, 10|17.1|43.2|57.1|23.2|52.8|65.7|43.2||

---

> ### Author Response · Authors · 2025-11-19
> **Response 2 - Q2**
>
> **Q2 (Weaknesses 2): Proposition 1 assumes second-order smoothness, which often doesn’t hold for large Transformers. How is this assumption justified in practice, and is there any empirical validation or relaxed version to support it?**
>
> **A2:**
> Thank you for raising this point. In Proposition 1, our second-order smoothness assumption states that the loss $\ell(\tilde{\mathcal{D}}, \theta)$ is twice differentiable with respect to the parameters $\theta$ along the considered trajectory, and that its Hessian varies in a locally Lipschitz way in this neighborhood. This assumption is only used to justify a second-order Taylor expansion with an $O({\Delta t}^2)$ remainder.
>
> In standard neural network architectures, all building blocks except the activation—linear projections, matrix multiplications, residual connections, softmax attention, layer normalization, and pooling—are smooth operations. We fully agree that a vanilla ReLU activation is not globally twice differentiable, and thus a pure ReLU network does not satisfy global second-order smoothness.
> However, most of today's mainstream network architectures (such as BERT, GPT, ViT, ConvNeXt, YOLOv5, EfficientNet) no longer use ReLU, but use smooth activation functions such as SiLU/GELU instead. Hence, in our main experiments, we **do not** use ReLU: all backbones (NFNet-L0, BERT, DINOv2/ViT, and BGE-1.5) employ the SiLU/GELU activation in their architectures, following the original implementations. SiLU/GELU is an infinitely differentiable smooth nonlinearity introduced precisely as a smoothed alternative to ReLU.
>
> Under these smooth activations, the loss is twice continuously differentiable with a locally Lipschitz Hessian on bounded parameter sets, so the second-order smoothness assumption of Proposition 1 is satisfied in our setting. This type of assumption is standard in recent analyses of second-order methods and Hessian properties for neural networks with smooth activations [1, 2], and our work follows this line of idealized modeling.
>
> We have not conducted a dedicated empirical study of Hessian Lipschitz constants, as such an investigation would be beyond the scope of this paper. Nevertheless, Figure 4 already shows that the theoretical bound from Proposition 1 closely tracks the empirical behavior of the shortcut trajectory, which supports the practical relevance of our smoothness assumption in the regimes considered here.
>
> *[1] First-order Stochastic Algorithms for Escaping from Saddle Points in Almost Linear Time, NeurIPS 2018.*
>
> *[2] Second-Order Optimization with Lazy Hessians, ICML 2023.*

---

> ### Author Response · Authors · 2025-11-19
> **Response 3 - Q3**
>
> **Q3 (Weaknesses 3): The method requires the full teacher training trajectory, which is costly. Can it still work with partial checkpoints or API-based teachers, or does it fundamentally depend on complete training?**
>
> **A3:**
> Thank you for this insightful comment. For large-scale teacher models with only partial checkpoints, our proposed distillation method can still be applied. However, for models accessible solely via an API, trajectory-matching approaches cannot be used since backward gradients cannot be computed. To address this concern, we first clarify our experimental setup and then describe additional methodology and experiments involving large teacher models.
>
> The goal of dataset distillation is to compress a large-scale dataset into a smaller synthetic dataset that can achieve comparable performance when training downstream models. Training a large teacher model from scratch is therefore unnecessary. In our experiments, we use **NFNet** as a trainable image encoder and **BERT** as a frozen text encoder, followed by a trainable projection head. Since BERT remains fixed during teacher training, this saves considerable computation.
>
> For larger baseline models without intermediate checkpoints, we can further **freeze both encoders** and train only a lightweight projection head. This allows us to save only the parameters of the projection head as the teacher trajectory. Consequently, even with access to only a pre-trained teacher model checkpoint, trajectory-matching distillation and our improvements can still be applied efficiently.
>
> To validate the practicality of this approach, we conduct experiments using larger backbones. Specifically, we employ frozen **DINO-v2** and **BGE-1.5** as the visual and textual encoders, respectively. The results demonstrate that our method remains effective even at these larger model scales.
>
> We have added these results to **Appendix D** and referenced them in **Section 4.1 (Main Results)** to further strengthen the empirical evidence.
>
> - *Results of DiNo-v2 and BGE-1.5 on COCO dataset. The model trained on the full dataset performs: IR@1=22.5, IR@5=50.8, IR@10=65.0, TR@1=31.7, TR@5=61.4, TR@10=74.0.*
>
> |pairs|method|IR@1|IR@5|IR@10|TR@1|TR@5|TR@10|mean|
> |-|-|-|-|-|-|-|-|-|
> |100|LoRS|5.8|20.6|30.7|10.2|29.4|39.8|22.7|
> ||ours|**8.1**|**23.5**|**34.9**|**13.5**|**33.8**|**46.4**|**26.7**|
> |200|LoRS|6.1|20.9|32.1|12.0|31.4|43.1|24.3|
> ||ours|**8.4**|**24.1**|**35.4**|**14.1**|**35.6**|**48.4**|**27.7**|
> |500|LoRS|7.7|22.9|33.3|12.7|33.5|46.1|26.0|
> ||ours|**9.1**|**25.9**|**37.2**|**15.0**|**36.5**|**50.1**|**29.0**|
>
> DiNo-v2: https://huggingface.co/timm/vit_base_patch16_224.dino
>
> BGE-1.5: https://huggingface.co/BAAI/bge-base-en-v1.5

---

> ### Author Response · Authors · 2025-11-26
>
> We are grateful to the reviewer for the insightful remarks. We have carefully responded to each of the points raised in the review and revised the manuscript accordingly. Should there be any remaining concerns that we have not fully resolved, we would very much welcome additional feedback and are glad to engage in further discussion.

---

### Official Review · Reviewer_Bjes · 2025-10-31

**Soundness:** 2
**Presentation:** 2
**Contribution:** 2
**Rating:** 2
**Confidence:** 4

**Summary:**

This paper presents a data distillation technique that aims to compress large multimodal datasets into a small set of synthetic samples that can train new models to achieve comparable performance for multimodal learning purposes. The authors propose the Phased Teacher Model with Shortcut Trajectory (PTM-ST) approach, which divides the training process into different phases by splitting the teacher trajectory and applying shortcut alignment between them, achieving a 13.5% improvement in retrieval metrics and an average gain of 9.53% on Flickr30K.

**Strengths:**

* Strong motivation. The authors explore phased knowledge gaps in multimodal dataset distillation by investigating the insight differences in distillation dynamics between multimodal and unimodal data distillation techniques, offering valuable finding into modality-specific learning behavior.

* Proposes a simple yet effective phased distillation strategy (PTM-ST), which divides the learning process into semantic phases and aligns them with the teacher’s learning dynamics for trajectory endpoint matching, improving overall distillation stability and performance.

**Weaknesses:**

- **Limited data scale and generalizability evidendence:** The experiments are conducted on small datasets (e.g., Flickr30K, COCO) with limited computational resources, which may not capture the knowledge-gap behavior at large scale. As a result, the observed early-phase effectiveness and multi-stage training strategy may not generalize to full-scale CLIP-style pretraining.

- **Additional training cost and model complexity:**  Requires more training and model complexity. Although the method aims to improve efficiency, it still requires fully training a large teacher model and dividing it into multiple phases, which substantially increases both computational and storage costs

- **Narrow experimental scope and weak comparison**: The experiments are confined to small benchmarks and a single teacher architecture (CLIP-base), without comparisons against stronger or larger vision–language models. This narrow setup limits the evidence supporting that “information-gap phasing” or session partitioning is a generally effective approach.

**Questions:**

It remains unclear how the proposed phased-teacher strategy generalizes under large-scale training. How do the observed “phase gaps” in representation or gradient dynamics behave when the dataset size grows substantially?

---

> ### Author Response · Authors · 2025-11-19
> **Response 1 - Q1, Q2**
>
> Thank you for your constructive comments! We give point-to-point replies to your questions in the following and hope these address your concerns.
>
> **Q1 (Weaknesses 1): Limited data scale and generalizability evidendence: The experiments use small datasets and limited compute, so they may not reflect large-scale behavior. Therefore, the early-phase effects and multi-stage strategy might not generalize to full-scale CLIP pretraining.**
>
> **A1:**
> Thank you for raising this concern. To further address it, we extend our method to a larger-scale dataset, LLaVA-CC3M, which contains 595k image–text pairs and is used for pre-training LLaVA. Following standard practice, we split the dataset into training, validation, and test subsets using a 3:1:1 ratio.
>
> Consistent with the analysis presented in Section 2.2, we first train the teacher model on this dataset to obtain its parameter trajectory and record performance along the trajectory. During distillation with 500 pairs, we then apply the original LoRS strategy, which naively shifts the trajectory-matching endpoint $T^+$ to later iterations. We measure the resulting student performance and compare it with the teacher.
>
> The results clearly show that the knowledge gap becomes even more pronounced at this larger scale. As training progresses, the teacher model continues to improve, whereas the student’s performance deteriorates substantially when following the shifted trajectory. This further validates our motivation for addressing the trajectory mismatch problem in large-scale settings.
>
>
> - *R-mean comparison between teacher and student on LLaVA-cc3m*
>
> |epoch / $T^+$|2|4|6|8|10|
> |-|-|-|-|-|-|
> |teacher|17.1|20.1|22.1|23.3|24.1|
> |student|7.2|4.4|2.9|2.5|2.7|
>
> Following this analysis, we further evaluate the effectiveness of our proposed method by conducting distillation experiments on the same large-scale dataset. As shown in the results table, our approach consistently alleviates the knowledge-gap issue and delivers improved distillation performance, even under the more challenging large-scale setting.
>
> We have incorporated these results into Section 4 of the paper to strengthen the overall empirical evidence.
>
> - *Distillation results on LLaVA-cc3m dataset. The model trained on the full dataset performs: IR@1=9.3, IR@5=25.9, IR@10=36.5, TR@1=9.8, TR@5=26.4, TR@10=37.3.*
>
> |pairs|method|IR@1|IR@5|IR@10|TR@1|TR@5|TR@10|mean|
> |-|-|-|-|-|-|-|-|-|
> |100|LoRS|1.2|4.6|7.7|1.7|6.9|11.4|5.6|
> ||ours|**2.3**|**8.2**|**13.2**|**2.9**|**10**|**15.9**|**8.8**|
> |200|LoRS|1.4|5.3|8.7|2.4|8.5|13.6|6.7|
> ||ours|**2.7**|**9.7**|**15.8**|**3.7**|**11.9**|**18.4**|**10.4**|
> |500|LoRS|1.7|6.2|10.1|2.5|8.7|13.8|7.2|
> ||ours|**3.2**|**10.4**|**16.6**|**3.9**|**12.8**|**19.7**|**11.1**|
> |1000|LoRS|1.6|5.5|8.9|2.7|9.1|15.1|7.2|
> ||ours|**3.3**|**11.4**|**17.9**|**4.1**|**13.2**|**19.9**|**11.6**|
>
> **Q2 (Weaknesses 2): Additional training cost and model complexity: Requires more training and model complexity. It still needs a fully trained large teacher and multiple phases, leading to higher compute and storage costs.**
>
> **A2:**
> Thank you for raising this concern. We apologize for the confusion: our method is **not** designed to improve training efficiency, and it does **not** introduce additional training overhead. Our clarification is as follows:
>
> 1. **No additional cost in obtaining the teacher trajectory.**
>    Our work follows prior trajectory-matching approaches (e.g., MTT-VL, LoRS), where training a teacher model to obtain its parameter trajectory is an essential and unavoidable step. We do *not* modify this stage in any way, and therefore incur **no extra training cost**.
>
> 2. **Computing interpolation-related distances is lightweight.**
>    To construct interpolated expert trajectories, we compute distances between parameter checkpoints along the teacher trajectory. This computation is performed **once**, typically completes within a few seconds, and is **negligible compared to the full distillation process**.
>
> 3. **Memory and storage overhead are not increased.**
>    Our method does not increase GPU memory consumption. In fact, due to the staged distillation design that splits a large synthetic dataset into smaller subsets, the GPU memory usage is **slightly reduced** (see Appendix D.6). For storage, we only save scalar distance values between checkpoints, which introduces **minimal disk overhead**.

---

> ### Author Response · Authors · 2025-11-19
> **Response 2 - Q3, Q4**
>
> **Q3 (Weaknesses 3): Narrow experimental scope and weak comparison: The experiments are confined to small benchmarks and a single teacher architecture (CLIP-base), without comparisons against stronger or larger vision–language models. This narrow setup limits the evidence supporting that “information-gap phasing” or session partitioning is a generally effective approach.**
>
> **A3:**
> Thank you for the constructive suggestion. To further validate the applicability of our method to larger-scale models, we conduct additional experiments using the more powerful **DINO-v2** and **BGE-1.5** as the image and text encoders, respectively. Both encoders were kept **frozen**, and a single-layer trainable projection head was added on top of them.
>
> Evaluation on the COCO dataset shows that **our method remains effective even when scaling up the model capacity**. This demonstrates that the proposed approach generalizes well across different backbone sizes.
>
> We have included this result in **Appendix D** and referenced it in **Section 4.1 (Main Results)** to further strengthen the empirical evidence.
>
> - *Results of DiNo-v2 and BGE-1.5 on COCO dataset. The model trained on the full dataset performs: IR@1=22.5, IR@5=50.8, IR@10=65.0, TR@1=31.7, TR@5=61.4, TR@10=74.0.*
>
> |pairs|method|IR@1|IR@5|IR@10|TR@1|TR@5|TR@10|mean|
> |-|-|-|-|-|-|-|-|-|
> |100|LoRS|5.8|20.6|30.7|10.2|29.4|39.8|22.7|
> ||ours|**8.1**|**23.5**|**34.9**|**13.5**|**33.8**|**46.4**|**26.7**|
> |200|LoRS|6.1|20.9|32.1|12.0|31.4|43.1|24.3|
> ||ours|**8.4**|**24.1**|**35.4**|**14.1**|**35.6**|**48.4**|**27.7**|
> |500|LoRS|7.7|22.9|33.3|12.7|33.5|46.1|26.0|
> ||ours|**9.1**|**25.9**|**37.2**|**15.0**|**36.5**|**50.1**|**29.0**|
>
>
> DiNo-v2: https://huggingface.co/timm/vit_base_patch16_224.dino
>
> BGE-1.5: https://huggingface.co/BAAI/bge-base-en-v1.5
>
> **Q4 (Questiones 1): It’s unclear whether the phased-teacher strategy holds at large scale—how “phase gaps” in representations or gradients change with much larger datasets remains unknown.**
>
> **A4:**
> Thank you for the question. To address this concern, we performed the gradient analysis described in Section 3.2 on the larger LLaVA-CC3M dataset. Specifically, we first train a teacher model on LLaVA-CC3M to obtain the **normal trajectory**, and then construct the **shortcut trajectory** by interpolating between the start and end points. We compute the gradient of the MTT loss with respect to the distillation dataset for both trajectories:
> $$
> \\nabla\_{\\tilde{\\mathcal{D}}}\\mathcal{L}\_{\\text{MTT}}(\\tilde{\\mathcal{D}}, \\theta\_T),\\quad \\mathcal{L}\_{\\text{MTT}}(\\tilde{\\mathcal{D}}, \\theta\_T)=\\frac{\\left\\Vert\\tilde{\\theta}\_{T+t}-\\theta\_{T+\\Delta T}\\right\\Vert\_2^2}{\\left\\Vert\\theta\_T-\\theta\_{T+\\Delta T}\\right\\Vert_2^2},\\ T=1,2,3,\\ldots
> $$
>
> where $\tilde{\theta}_{T+t}$ denotes the parameters obtained by training the model on the synthetic dataset $\tilde{\mathcal{D}}$ starting from $\theta_T$ for $t$ steps.
>
> Flattening these gradients and computing their cosine similarity, we observe that **under the original training trajectory**, the gradient similarity across different epochs is generally low, indicating significant gaps between teacher parameters at different training stages. In contrast, **our proposed shortcut strategy** substantially increases gradient similarity across stages, resulting in a more consistent and continuous distillation process.
>
> These results demonstrate that our method effectively mitigates the phase gap problem. More importantly, the approach shows stable and superior performance not only on the original Flickr30k and MS-COCO datasets, but also on the larger LLaVA-CC3M dataset, further validating its generality and robustness.
>
> - *Gradient cosine similarity (normal)*
>
> |$T$|1|2|3|4|5|6|7|8|9|
> |-|-|-|-|-|-|-|-|-|-|
> |1|1.00|0.08|0.11|0.02|-0.03|-0.04|-0.05|-0.06|-0.05|
> |2|0.08|1.00|0.07|0.26|0.24|0.18|0.15|0.11|0.09|
> |3|0.11|0.07|1.00|0.40|0.55|0.50|0.44|0.40|0.34|
> |4|0.02|0.26|0.40|1.00|0.66|0.71|0.66|0.59|0.55|
> |5|-0.03|0.24|0.55|0.66|1.00|0.79|0.80|0.76|0.71|
> |6|-0.04|0.18|0.50|0.71|0.79|1.00|0.86|0.86|0.82|
> |7|-0.05|0.15|0.44|0.66|0.80|0.86|1.00|0.89|0.90|
> |8|-0.06|0.11|0.40|0.59|0.76|0.86|0.89|1.00|0.91|
> |9|-0.05|0.09|0.34|0.55|0.71|0.82|0.90|0.91|1.00|
>
> - *Gradient cosine similarity (shortcut)*
>
> |$T$|1|2|3|4|5|6|7|8|9|
> |-|-|-|-|-|-|-|-|-|-|
> |1|1.00|0.85|0.69|0.59|0.53|0.50|0.47|0.45|0.43|
> |2|0.85|1.00|0.94|0.89|0.84|0.81|0.78|0.75|0.73|
> |3|0.69|0.94|1.00|0.98|0.95|0.93|0.90|0.88|0.86|
> |4|0.59|0.89|0.98|1.00|0.99|0.97|0.96|0.94|0.92|
> |5|0.53|0.84|0.95|0.99|1.00|0.99|0.98|0.97|0.95|
> |6|0.50|0.81|0.93|0.97|0.99|1.00|0.99|0.98|0.97|
> |7|0.47|0.78|0.90|0.96|0.98|0.99|1.00|0.99|0.99|
> |8|0.45|0.75|0.88|0.94|0.97|0.98|0.99|1.00|0.99|
> |9|0.43|0.73|0.86|0.92|0.95|0.97|0.99|0.99|1.00|

---

> > ### Comment · Reviewer_Bjes · 2025-11-25
> >
> > Thanks for the detailed and constructive author feedbacks. After reviewing the clarifications, the additional experiments, I have updated my score accordingly.

---

> ### Author Response · Authors · 2025-11-25
>
> We sincerely thank you for your careful reading and thoughtful feedback. We are encouraged by your acknowledgement.
>
> If you have any additional comments or unresolved concerns, please let us know. We are happy to provide further clarification.

---

### Official Review · Reviewer_nf5n · 2025-11-01

**Soundness:** 3
**Presentation:** 2
**Contribution:** 3
**Rating:** 6
**Confidence:** 3

**Summary:**

- The paper addresses dataset distillation, a key topic for enabling fast and cost-effective training of student models.
- It proposes a novel dataset distillation method specifically designed to be effective in multimodal settings.
- The core contribution is using intermediate checkpoints from a teacher model to distill datasets that capture the teacher's learning trajectory.
- The method utilizes sets of teacher checkpoints to create the distilled dataset, rather than sampling from individual steps, to provide a more global view.

**Strengths:**

- The method demonstrates promising empirical results, achieving strong and consistent improvements over the compared baselines.
- The approach of using sets of teacher parameters (checkpoints) effectively provides the student model with a global perspective on the teacher's training dynamics.

**Weaknesses:**

- The method requires access to intermediate teacher checkpoints. For very large models (the bigger CLIP variants), these are often unavailable, and replicating the teacher training to generate them introduces significant computational overhead.
- The student model's architecture and specific training details are missing.

**Questions:**

See above.

---

> ### Author Response · Authors · 2025-11-19
>
> We sincerely thank the reviewer for the insightful suggestions on both the content and experimental aspects of our paper. In response, we carefully addressed each point in the replies below.
>
> **Q1 (Weaknesses 1): The method depends on intermediate teacher checkpoints, which are often unavailable for large models. Reproducing them requires retraining the teacher and adds substantial computational cost.**
>
> **A1:**
> We thank the reviewer for raising this concern. To address it, we first clarify our experimental setup and then provide additional methodology and experiments involving larger teacher models.
>
> The goal of dataset distillation is to compress a large-scale dataset into a significantly smaller synthetic dataset that can still yield comparable performance when training downstream models. Therefore, training an extremely large teacher model from scratch is unnecessary. In our experiments, we adopt NFNet as the trainable image encoder, BERT as a frozen text encoder, followed by a trainable projection head. During teacher training, BERT remains fixed, which reduces the computational cost.
>
> For larger teacher models for which full training checkpoints may not be available, we further freeze the image encoder and train only an additional projection head. To validate the feasibility of this strategy, we conduct experiments using larger backbones—specifically, frozen DINO-v2 for the visual modality and frozen BGE-1.5 for the textual modality. The results show that our method remains effective even at substantially larger model scales.
>
> We have added this set of results to Appendix D and referenced it in the main results section (Section 4.1) to strengthen the clarity and persuasiveness of our methodology.
>
> - *Results of DiNo-v2 and BGE-1.5 on COCO dataset. The model trained on the full dataset performs: IR@1=22.5, IR@5=50.8, IR@10=65.0, TR@1=31.7, TR@5=61.4, TR@10=74.0.*
>
> |pairs|method|IR@1|IR@5|IR@10|TR@1|TR@5|TR@10|mean|
> |-|-|-|-|-|-|-|-|-|
> |100|LoRS|5.8|20.6|30.7|10.2|29.4|39.8|22.7|
> ||ours|**8.1**|**23.5**|**34.9**|**13.5**|**33.8**|**46.4**|**26.7**|
> |200|LoRS|6.1|20.9|32.1|12.0|31.4|43.1|24.3|
> ||ours|**8.4**|**24.1**|**35.4**|**14.1**|**35.6**|**48.4**|**27.7**|
> |500|LoRS|7.7|22.9|33.3|12.7|33.5|46.1|26.0|
> ||ours|**9.1**|**25.9**|**37.2**|**15.0**|**36.5**|**50.1**|**29.0**|
>
> DiNo-v2: https://huggingface.co/timm/vit_base_patch16_224.dino
>
> BGE-1.5: https://huggingface.co/BAAI/bge-base-en-v1.5
>
> **Q2 (Weaknesses 2): The student model's architecture and specific training details are missing.**
>
> **A2:**
> Thanks to the reviewer for the reminder. In our main experiments, the student model shares the same architecture as the teacher model, i.e., NFNet + BERT. We train the student model using the SGD optimizer with momentum set to 0.9 and weight decay set to 0.0005. The remaining hyperparameters are listed in the table.
>
> It is worth noting that during the distillation process, the learning rate for training the student model is itself learnable. The value reported in the table corresponds to its initialization, with the initial learning-rate learning-rate set to `lr_lr = 0.01`.
>
> We have added the full training details of the student model to Appendix D.1 of the paper.
>
> - *Hyperparameter settings for student training*
>
> ||Flickr30K|MS-COCO|
> |-|-|-|
> |epoch(per subset)|50|50|
> |batch_size|128|128|
> |lr_img|0.1|0.1|
> |lr_txt|0.1|0.1|
> |image_size|224×224|224×224|

---

> ### Author Response · Authors · 2025-11-26
>
> We thank the reviewer once more for the thoughtful feedback. We have carefully responded to each of the points raised in the review and revised the manuscript accordingly. If any aspects still appear unclear or unresolved, we would be very grateful to hear your further thoughts and are happy to clarify anything that remains.

---

> > ### Comment · Reviewer_nf5n · 2025-11-27
> >
> > I thank the authors for clarifying that the method works with training a projection layer on the teacher model, thereby removing the need for intermediate checkpoints. This addresses my concern regarding this constraint.

---

> > > ### Author Response · Authors · 2025-11-27
> > >
> > > Thank you very much for your follow-up and for clarifying that this addresses your concern. If there are any other aspects that remain unclear or any additional questions arise, we would be very happy to provide further clarification.

---

### Official Review · Reviewer_dydj · 2025-11-02

**Soundness:** 3
**Presentation:** 3
**Contribution:** 3
**Rating:** 6
**Confidence:** 2

**Summary:**

This paper identifies a issue called "phased knowledge gap" in multimodal dataset distillation where student models fail to learn effectively from teacher models in the later stages of training. This paper then proposes Phased Teacher Model with Shortcut Trajectory (PTM-ST) to address this challenge. The core idea is to decompose the distillation process into phases, using different teacher models at each stage to provide more stable guidance. Additionally, it introduces a "Shortcut Trajectory" to create a smoothed and stabilized learning path for the student model. The proposed method is evaluated on Flickr30K and COCO datasets for image-text retrieval, demonstrating significant improvements over existing state-of-the-art methods.

**Strengths:**

- Novelty. The paper identifies the "phased knowledge gap" as a critical issue in multimodal dataset distillation, where student performance degrades when using teacher models from later training stages. The proposed PTM-ST framework is a novel and effective solution that addresses this problem through phased learning and trajectory stabilization.

- Strong empirical performance. The experimental results on Flickr30K and COCO datasets show that PTM-ST consistently and significantly outperforms existing baselines across all metrics.

- Thorough analysis and ablations. The paper provides a comprehensive analysis of the problem, supported by visualizations of gradient instability and theoretical arguments for the proposed solution's stability. The ablation studies in Tables 3 effectively demonstrate the contribution of each component of the PTM-ST framework (PTM, ST, and EMA) to the overall performance improvement.

**Weaknesses:**

- The PTM-ST framework introduces additional complexity, requiring manual specification of interpolation endpoints and matching ranges for each distillation stage. This may make the method difficult to apply to new datasets or tasks and raises concerns about hyperparameter sensitivity.

- Limited Scope of Evaluation: The experiments are limited to image-text retrieval. It would be beneficial to evaluate the generalizability of PTM-ST on other multimodal tasks, such as Visual Question Answering (VQA) or image captioning.

**Questions:**

The authors are encouraged to compare with the following NeurIPS 2025 paper.

Efficient Multimodal Dataset Distillation via Generative Models.

---

> ### Author Response · Authors · 2025-11-19
> **Response 1 - Q1, Q2**
>
> Thank you for your time and constructive comments. We have responded to all your questions as follows and hope these address your concerns.
>
> **Q1 (Weaknesses 1): The PTM-ST framework adds extra complexity because it needs manually chosen interpolation endpoints and matching ranges for each distillation stage. This makes it harder to use on new datasets or tasks and increases sensitivity to hyperparameters.**
>
> **A1:**
> Thank you for raising this point. In summary, while our method does require specifying these parameters, we find that the performance is not sensitive to their values. In Appendix D.3, we analyze the impact of different interpolation endpoints on distillation performance. For the matching range $(T_p^-, T_p^+)$, we adopt a simple linear strategy in our experiments: $T_p^-=T_1^-+\Delta d(p-1)$, $T_p^+=T_1^++\Delta d(p-1)$. To further assess the effect of varying $\Delta d$, we conduct additional ablation studies under the Flickr30k 500-pair setting.
>
> The results show that although different parameter choices introduce some variation, the overall influence is limited; more importantly, our method consistently outperforms the LoRS baseline across all configurations. This demonstrates the robustness and practicality of our approach. We have added these comparison results to Appendix D.3 to ensure completeness.
>
> - *Appendix D.3: Comparison of different interpolation endpoints.*
>
> |Pairs|Interpolation Endpoint|IR@1|IR@5|IR@10|TR@1|TR@5|TR@10|Mean|LoRS|
> |-|-|-|-|-|-|-|-|-|-|
> |100|6|9.6|28.4|41.5|14.4|38.6|52.7|30.9|25.6|
> ||8|8.4|26.6|39.7|15.4|39.9|54.6|30.8||
> ||10|8.8|26.8|39.7|14.9|39.6|53.6|30.6||
> |200|4,6|11.7|33.2|46.3|18.5|43.5|58.6|35.3|30.1|
> ||6,8|12.5|34.7|48.5|19.3|45.9|58.9|36.6||
> ||8,10|11.2|32.3|45.4|18.1|46.4|60.2|35.6||
> |500|4,6|15.3|39.6|53.1|22.6|49.2|63.4|40.5|32.3|
> ||6,8|16.0|40.5|54.0|22.2|51.1|64.6|41.4||
> ||8,10|14.9|38.8|52.2|22.1|51.2|64.4|40.6||
> |1000|4,7,10|16.6|42.5|56.2|23.7|50.2|62.5|42.0|31.4|
> ||6,8,10|17.5|42.8|56.3|23.8|53.6|67.2|43.5||
> ||8,9,10|17.1|43.2|57.1|23.2|52.8|65.7|43.2||
>
> - *Comparison of different matching ranges.*
>
> |Method|IR@1|IR@5|IR@10|TR@1|TR@5|TR@10|mean|
> |-|-|-|-|-|-|-|-|
> |$\Delta d=0$|15.2|39.6|53.1|21.6|48.6|62.9|40.2|
> |$\Delta d=1$|16.0|40.5|54.0|22.2|51.1|64.6|41.4|
> |$\Delta d=2$|15.9|40.1|53.2|21.0|50.1|63.7|40.7|
> |LoRS|12.7|32.9|44.9|14.7|37.6|51.1|32.3|
>
> **Q2 (Weaknesses 2): Limited Scope of Evaluation: The evaluation is restricted to image-text retrieval, so it's unclear how well PTM-ST generalizes to other multimodal tasks like VQA or image captioning.**
>
> **A2:**
> Thanks for your constructive suggestion. First, to avoid potential misunderstanding, we would like to clarify that—consistent with the prior work LoRS—our model architecture follows a CLIP-style architecture and therefore does not support the image captioning task. To prevent confusion, we have revised the description in the third paragraph of Section 4.1 to make this point clearer.
>
> To more comprehensively evaluate the generalization capability of our distillation method, we further conduct experiments on Visual Question Answering (COCO-QA [1]) and ImageNet classification. Specifically, in COCO-QA, we enumerate all possible answers and construct the template "Question: {question} Answer: {answer}." for each question–answer pair, and then compute the Top-K retrieval performance. For ImageNet, we randomly sample 50 categories from ImageNet-1K and report the Top-K zero-shot classification accuracies.
>
> In these comparisons, teacher denotes the model trained on the full COCO dataset and evaluated directly in a zero-shot manner on both tasks. In contrast, LoRS and Ours are trained under the same distillation budget, using only 499 COCO image–text pairs. As summarized in the table, under identical supervision budgets, our method significantly outperforms LoRS and further narrows the performance gap relative to the full-data teacher model.
>
> We have incorporated these experiments into Section 4 of the revised manuscript to strengthen the completeness and persuasiveness of our evaluation.
>
> - *Visual Question Answering results on COCO-QA*
>
> |Methods|ACC1|ACC5|ACC10|
> |-|-|-|-|
> |teacher|26.3|48|55.5|
> |LoRS|10.8|33|39.9|
> |ours|**16.3**|**38.9**|**47.4**|
>
>
> - *ImageNet-50 classification*
>
> |Methods|ACC1|ACC5|ACC10|
> |-|-|-|-|
> |teacher|29.6|59.3|73.5|
> |LoRS|18.6|42.1|55.9|
> |ours|**22.1**|**52.7**|**67.9**|
>
> COCO-QA: http://www.cs.toronto.edu/~mren/imageqa/data/cocoqa/cocoqa-2015-05-17.zip
>
> [1] Hierarchical question-image co-attention for visual question answering. NeurIPS, 2016.

---

> ### Author Response · Authors · 2025-11-19
> **Response 2 - Q3**
>
> **Q3 (Questions 1): The authors are encouraged to compare with the following NeurIPS 2025 paper. Efficient Multimodal Dataset Distillation via Generative Models.**
>
> **A3:**
> We appreciate your constructive recommendation. We provide the comparative results on both Flickr30k and MS-COCO in the table below. As shown, our method consistently outperforms both LoRS and EDGE across the two datasets. These results demonstrate the robustness and strong generalization ability of our approach under varying dataset sizes and distributions.
>
> In contrast, EDGE primarily focuses on reducing memory consumption through diffusion-based techniques, thereby improving the feasibility of large-scale distillation. To further evaluate our method in a more challenging large-scale setting, we adopt the LLaVA-CC3M dataset, which contains 595k image–text pairs used for LLaVA pre-training. We split this dataset into training, validation, and test sets following a 3:1:1 ratio.
>
> We have incorporated these comparisons and the additional experiments into Section 4 of the revised manuscript to strengthen the comprehensiveness of our evaluation.
>
>
> - *Results of 500 pairs on Flickr30K and COCO dataset.*
>
> |Dataset|Metric|LoRS|EDGE|Ours|
> |-|-|-|-|-|
> |Flickr30K|IR@1|12.7|6.7|**16.0**|
> ||IR@5|32.9|21.0|**40.5**|
> ||IR@10|44.9|30.5|**54.0**|
> ||TR@1|14.7|13.3|**22.2**|
> ||TR@5|37.6|35.6|**51.1**|
> ||TR@10|51.1|47.5|**64.6**|
> ||Mean|32.3|25.8|**41.4**|
> |COCO|IR@1|3.3|1.8|**6.6**|
> ||IR@5|11.8|6.5|**20.5**|
> ||IR@10|19.2|11.2|**30.7**|
> ||TR@1|4.1|2.9|**6.9**|
> ||TR@5|12.8|9.5|**20.1**|
> ||TR@10|20.2|15.7|**30.0**|
> ||Mean|11.9|7.9|**19.1**|
>
> - *Results of 1000 pairs on Flickr30K and COCO dataset.*
>
> |Dataset|Metric|LoRS|EDGE|Ours|
> |-|-|-|-|-|
> |Flickr30K|IR@1|9.8|9.9|**17.5**|
> ||IR@5|29.0|28.2|**42.8**|
> ||IR@10|41.6|40.5|**56.3**|
> ||TR@1|14.9|14.5|**23.8**|
> ||TR@5|39.8|38.3|**53.6**|
> ||TR@10|53.5|51.7|**67.2**|
> |COCO|IR@1|2.5|2.8|**7.0**|
> ||IR@5|9.9|9.8|**21.8**|
> ||IR@10|16.6|16.2|**32.3**|
> ||TR@1|4.2|3.9|**6.8**|
> ||TR@5|15.2|13.0|**20.9**|
> ||TR@10|24.1|21.0|**30.8**|
>
>
> - *Results on LLaVA-CC3M dataset. The model trained on the full dataset performs: IR@1=9.3, IR@5=25.9, IR@10=36.5, TR@1=9.8, TR@5=26.4, TR@10=37.3.*
>
> |pairs|method|IR@1|IR@5|IR@10|TR@1|TR@5|TR@10|mean|
> |-|-|-|-|-|-|-|-|-|
> |100|LoRS|1.2|4.6|7.7|1.7|6.9|11.4|5.6|
> ||ours|**2.3**|**8.2**|**13.2**|**2.9**|**10**|**15.9**|**8.8**|
> |200|LoRS|1.4|5.3|8.7|2.4|8.5|13.6|6.7|
> ||ours|**2.7**|**9.7**|**15.8**|**3.7**|**11.9**|**18.4**|**10.4**|
> |500|LoRS|1.7|6.2|10.1|2.5|8.7|13.8|7.2|
> ||ours|**3.2**|**10.4**|**16.6**|**3.9**|**12.8**|**19.7**|**11.1**|
> |1000(LLaVA-cc3m)|LoRS|1.6|5.5|8.9|2.7|9.1|15.1|7.1|
> ||ours|**3.3**|**11.4**|**17.9**|**4.1**|**13.2**|**19.9**|**11.6**|
> |1000(CC3M)|EDGE|0.2|0.5|1.0|0.1|0.7|1.1|0.6|

---

> ### Author Response · Authors · 2025-11-26
>
> We would like to thank the reviewer again for the constructive comments. We have carefully responded to each of the points raised in the review and revised the manuscript accordingly. If there are any remaining concerns or issues that we have not fully addressed, we would greatly appreciate further feedback and would be happy to continue the discussion.

---

### Author Response · Authors · 2025-12-01
**Summary of Rebuttal**

Dear Area Chairs (ACs),

We sincerely appreciate your time and effort in reviewing our paper.

In response to the reviewers’ insightful comments, we have made substantial revisions and additions to the manuscript. We conducted new experiments and analyses directly targeting the core concerns and provided clearer, more detailed explanations of the experimental design and results in the main text.

We are encouraged that several reviewers explicitly noted that the revised manuscript successfully addresses their concerns and acknowledged the improvements. Overall, we believe the updated version significantly strengthens the clarity, rigor, and technical depth of our work.

| Reviewer | Initial Rating | Confidence | Post-rebuttal Feedback |
| :--- | :--- | :--- | :--- |
| dydj | 6 | 2 | None |
| nf5n | 6 | 3 | **Positive** |
| Bjes | 2 | 4 | **Positive** |
| SUGG | 4 | 3 | None |

**Note:** 'None' denotes that we've not received feedback from the review until now.

Best regards,

The authors

---

> ### Author Response · Authors · 2025-12-01
> **Summary of Rebuttal (Part 1/4): Reviewer dydj (Initial Rating: 6, Confidence: 2, Feedback: None)**
>
> Reviewer dydj praised the novelty of identifying the "phased knowledge gap" and the method's strong empirical performance, but raised concerns about hyperparameter sensitivity, limited evaluation tasks, and missing comparisons.
>
> > **Concern 1 (Complexity):** Hyperparameter sensitivity (interpolation endpoints/matching ranges).
> *   **Solution:** We clarified the simple linear strategy used and provided ablation studies on endpoints and matching ranges ($\Delta d$) in **Appendix D.3**.
> *   **Conclusion:** The method is robust to parameter variations and consistently outperforms baselines across configurations.
>
> > **Concern 2 (Scope):** Limited evaluation (only retrieval); requested generalization tests on VQA/Captioning.
> *   **Solution:** We extended evaluations to **Visual Question Answering (COCO-QA)** and **ImageNet-50 Zero-shot Classification** (added to **Section 4**), clarifying architectural limitations for captioning.
> *   **Conclusion:** PTM-ST significantly outperforms LoRS on both VQA and classification, demonstrating strong generalization.
>
> > **Concern 3 (Comparison):** Requested comparison with NeurIPS 2025 paper (EDGE).
> *   **Solution:** We compared our method against EDGE on Flickr30K, MS-COCO, and LLaVA-CC3M datasets (added to **Section 4**).
> *   **Conclusion:** Our method consistently outperforms both LoRS and EDGE across all datasets, delivering superior distillation quality.
>
> **Outcome:**
> We verified hyperparameter robustness, extended evaluations to VQA/Classification, and outperformed the suggested method (EDGE) to address the raised concerns. We have updated the manuscript with these results and believe that our responses have addressed the reviewer‘s concerns.

---

> ### Author Response · Authors · 2025-12-01
> **Summary of Rebuttal (Part 2/4): Reviewer nf5n (Initial Rating: 6, Confidence: 3, Feedback: Positive)**
>
> Reviewer nf5n recognized the effectiveness of using teacher checkpoint sets but questioned the practicality of obtaining checkpoints for large models and noted missing student details.
>
> > **Concern 1 (Practicality):** Difficulty and cost of obtaining intermediate checkpoints for large models.
> *   **Solution:** We used a **"Frozen Encoder + Trainable Projector"** strategy to avoid full retraining and validated it with **frozen DINO-v2 and BGE-1.5** in **Appendix D.4**.
> *   **Conclusion:** Our method is applicable to large foundation models. The results show that our method remains effective at substantially larger model scales.
>
> > **Concern 2 (Reproducibility):** Missing student model architecture and training details.
> *   **Solution:** We added full details (NFNet+BERT architecture, SGD optimizer, learnable LR mechanism) to **Appendix D.1**.
> *   **Conclusion:** The method is now fully reproducible.
>
> **Outcome:**
> The reviewer explicitly commented: **"This addresses my concern regarding this constraint."**

---

> ### Author Response · Authors · 2025-12-01
> **Summary of Rebuttal (Part 3/4): Reviewer Bjes (Initial Rating: 2, Confidence: 4, Feedback: Positive)**
>
> Reviewer Bjes praised the motivation but raised concerns regarding data scale/generalizability, training costs, and the narrow scope of architectures.
>
> > **Concern 1 (Data Scale):** Limited evidence on large-scale generalizability; questioned if "knowledge gaps" persist.
> *   **Solution:** We extended the method to **LLaVA-CC3M (595k pairs)** in **Section 4**, confirming that trajectory mismatch worsens at scale, and PTM-ST effectively mitigates it.
> *   **Conclusion:** The strategy generalizes effectively to large-scale pretraining datasets.
>
> > **Concern 2 (Cost):** Potential additional training cost and model complexity.
> *   **Solution:** We clarified that teacher training is standard (no extra cost) and demonstrated in **Appendix D.6** that staged distillation actually reduces peak GPU memory.
> *   **Conclusion:** Improved stability is achieved without significant computational or storage overhead.
>
> > **Concern 3 (Scope):** Narrow experimental scope; require comparison with stronger Vision-Language Models.
> *   **Solution:** We validated the method on stronger backbones (**DINO-v2** and **BGE-1.5**) on COCO (results in **Appendix D.4**).
> *   **Conclusion:** The method remains effective across diverse and stronger architectures.
>
> > **Concern 4 (Analysis):** Unclear gradient dynamics ("phase gaps") at scale.
> *   **Solution:** We performed gradient similarity analysis on **LLaVA-CC3M**, showing our shortcut strategy significantly improves gradient alignment over normal trajectories.
> *   **Conclusion:** Empirically validated the mechanism on large-scale data.
>
> **Outcome:**
> Although reviewer Bjes initially assigned a lower score, he explicitly acknowledged that the rebuttal **addressed these concerns and clearly indicated an upward adjustment**.

---

> ### Author Response · Authors · 2025-12-01
> **Summary of Rebuttal (Part 4/4): Reviewer SUGG (Initial Rating: 4, Confidence: 3, Feedback: None)**
>
> Reviewer SUGG valued the systematic analysis of "phase-wise knowledge drift" and the performance improvement but questioned hyperparameter sensitivity, theoretical assumptions, and costs.
>
> > **Concern 1 (Sensitivity):** Sensitivity to match intervals / endpoints; suggested adaptive tuning.
> *   **Solution:** We conducted ablation studies on matching ranges and interpolation endpoints in **Appendix D.3**, showing stable performance.
> *   **Conclusion:** The method is robust to hyperparameter choices. We will explore the possibility of adaptive parameter selection.
>
> > **Concern 2 (Theory):** Validity of "second-order smoothness" assumption for Transformers.
> *   **Solution:** We clarified that our architectures (BERT, ViT, NFNet) use smooth **GELU/SiLU** activations, satisfying the assumption.
> *   **Conclusion:** The assumption is theoretically valid and empirically supported by trajectory analysis (**Figure 4**).
>
> > **Concern 3 (Cost):** Requirement for full teacher training trajectory limits applicability.
> *   **Solution:** We used the **"Frozen Encoder"** strategy (only training projector) for large models and validated our method on stronger backbones (DINO-v2 and BGE-1.5) on COCO.
> *   **Conclusion:** Enables efficient application to large models without full retraining.
>
> **Outcome:**
> We conducted new experiments and clarified theoretical assumptions to respond to these concerns. We have revised the manuscript accordingly and believe that our responses have addressed the reviewers’ concerns.

---

### Meta-Review · Area_Chair_7gN7 · 2026-01-10

**Summary:**

The paper proposed Phased Teacher Model with Shortcut Trajectory (PTM-ST) to address the limitations of existing multimodal dataset distillation methods. It identifies the “phased knowledge gap” problem tthat student models struggle to learn from teacher models in later training stages, and proposes a phased distillation strategy that models teacher dynamics across training phases. The method incorporates shortcut trajectories to stabilize optimization and improve knowledge transfer. Empirical results on Flickr30K, COCO, and LLaVA-CC3M demonstrate improvements over state-of-the-art baselines.

**Reviewer Concerns:**

Reviewers raised several concerns. Reviewer dydj questioned the method’s sensitivity to hyperparameters and its limited evaluation scope. Reviewer nf5n highlighted the practical challenge of obtaining intermediate teacher checkpoints for large models and noted missing student training details. Reviewer Bjes concerns about the method’s scalability, training cost, and generalizability to stronger architectures. Reviewer SUGG echoed concerns about hyperparameter tuning, theoretical assumptions, and the cost of full teacher trajectories. Despite these concerns, most reviewers acknowledged the novelty and empirical strength of the proposed approach.

**Reviewer Scores:**

The paper received mixed but generally positive scores after rebuttal. Reviewer dydj and nf5n both rated it a 6, and nf5n explicitly stating that post-rebuttal clarifications addressed concerns. Reviewer SUGG gave a 4, did not provide after rebuttal feedback, while reviewer Bjes initially gave a 2 but later acknowledged that the authors’ rebuttal addressed key issues, suggesting a possible score revision. Overall, the paper was seen as a promising contribution with strong empirical results, though concerns about complexity and generalizability remain. The authors should carefully revise the final version, especially the discussion regarding the practicality of requiring intermediate teacher checkpoints under different deployment scenarios.

---

### Decision · Program_Chairs · 2026-01-26

Accept (Poster)